# When less is more powerful: Shapley value attributed ablation with augmented learning for practical time series sensor data classification

**Arijit Ukil**[1]*, **Leandro Marin**[2], **Antonio J. Jara**[3]

**1** TCS Research, Tata Consultancy Services, Kolkata, India, **2** Faculty of Computer Science, University of Murcia, Murcia, Spain, **3** Libelium, Murcia, Spain

* arijit.ukil@tcs.com

**Data Availability Statement:** All relevant data is available from the figshare repository (https://doi.org/10.6084/m9.figshare.21532440.v2).

**Funding:** The work is partially funded by Grant PID2020-112675RB-C44 funded by MCIN

## Abstract

Time series sensor data classification tasks often suffer from training data scarcity issue due to the expenses associated with the expert-intervened annotation efforts. For example, Electrocardiogram (ECG) data classification for cardio-vascular disease (CVD) detection requires expensive labeling procedures with the help of cardiologists. Current state-of-the-art algorithms like deep learning models have shown outstanding performance under the general requirement of availability of large set of training examples. In this paper, we propose Shapley Attributed Ablation with Augmented Learning: ShapAAL, which demonstrates that deep learning algorithm with suitably selected subset of the seen examples or ablating the unimportant ones from the given limited training dataset can ensure consistently better classification performance under augmented training. In ShapAAL, additive perturbed training augments the input space to compensate the scarcity in training examples using Residual Network (ResNet) architecture through perturbation-induced inputs, while Shapley attribution seeks the subset from the augmented training space for better learnability with the goal of better general predictive performance, thanks to the "efficiency" and "null player" axioms of transferable utility games upon which Shapley value game is formulated. In ShapAAL, the subset of training examples that contribute positively to a supervised learning setup is derived from the notion of coalition games using Shapley values associated with each of the given inputs' contribution into the model prediction. ShapAAL is a novel push-pull deep architecture where the subset selection through Shapley value attribution pushes the model to lower dimension while augmented training augments the learning capability of the model over unseen data. We perform ablation study to provide the empirical evidence of our claim and we show that proposed ShapAAL method consistently outperforms the current baselines and state-of-the-art algorithms for time series sensor data classification tasks from publicly available UCR time series archive that includes different practical important problems like detection of CVDs from ECG data.

(Ministry for Science and Innovation)/AEI (Agencia Estatal de Investigación - State Research Agency)/ 10.13039/5011000011033. Tata Consultancy Services is funding the work with generous support and entire APC funding.

**Competing interests:** The authors have declared that no competing interests exist.

## Introduction

With the advent of Internet of Things (IoT) and ever-increasing adoptions of sensors in the physical world, analytics problems with practical relevance are growing in numbers. One of the typical real-world challenges is to solve different classification problems, particularly that deal with time series sensor data to build sensing intelligence as one of the most useful practical implementations of artificial intelligent (AI) technique. We like to acknowledge the capability of the remarkably improved deep learning algorithms powered by the computational strength of high-powered computing infrastructure including different cloud platforms and Graphics Processing Unit (GPU)-based servers and work-stations [1, 2]. The ubiquity of smartphones and smart devices including smart bands, smart watches, smart gears, and the development of advanced sensors are playing an important role to leverage the substantial improvement in the sensing technologies to capture the physical and physiological information. High end GPU-enabled computing, cloud infrastructure, public availability of useful data sources and the emergence of powerful AI techniques like deep learning algorithms pose us the opportunity of developing vast number of worthy applications [3–5]. Currently, we are witnessing the learning revolution paradigm, where providing examples or training instances are often sufficient for a machine or computer to learn substantially such that it can be comparable to human-level ability. Sensors capture the physical world information from its ambience and provide the required inputs to the intelligent system such that it can sense the given physical space and perform different decision-making processes. Sensors can be considered as the micro-representation of our physical and physiological spaces.

The fundamental focus of this work is to find solutions of such practical yet diverse real world problems. Time series data are omnipresent in large set of practical applications, especially where sensor data are used to build intelligent systems. Sensors like Electrocardiogram (ECG), accelerometer, Infra-red spectroscopy, smart electric meter, etc. generate time series outputs, which motivate us such to build reliable time series classification model. For example, the important problems like cardio-vascular disease conditions like Atrial Fibrillation detection [6, 7] or Myocardial Infarction (commonly known as heart attack condition) detection from ECG data are of immense practical importance [8]. However, real-world problems come with different types of practical challenges. We particularly consider the time series sensor data classification problem, where the task is to build multi-class classification models by training given time series sensor data. We observe that large set of real-world time series sensor datasets ([8]) often suffer from the scarcity in labeled training examples for various reasons like expensive process of experimental setup or limited availability of the experimental setup (for e.g., "SonyAIBORobotSurface1" dataset [8] requires a robot to walk on different kinds of surfaces like cement or carpet) as well as the expenses associate with annotation process ("ECG200" dataset [8] requires cardiologists to annotate data whether the ECG recording is a normal sinus rhythm or Myocardial Infarction condition). "SonyAIBORobotSurface1" dataset contains mere 20 number of training examples, "ECG200" contains 100 number of training examples. Traditionally, deep neural networks require large set of training datasets for reliable and generalized learning. For example, CIFAR-10 dataset consists of 50,000 training images, while classical ImageNet 2012 classification dataset consists 1.28 million training datasets [9, 10]. CIFAR-10 and CIFAR-100 are actually labeled subsets of 80 million image [11]. Such abundance of training dataset availability is infeasible in case of practical time series sensor signal analysis problems. In fact, deep learning algorithms rely on the sufficiency of the training examples with the assumption that the learned embeddings preserve latent structures and the distribution of the given time series data [12].

Typically the solution of training data limitation is tackled by augmented learning through adversarial training [13–15], where the input training space is augmented through perturbation. Adversarial examples, which in simple terms are the perturbed forms of the input training data, have potential benefit as a data augmentation method to solve the training data scarcity issue [16, 17]. However, adversarial examples need finer control and it is shown that adversarial training mostly positively helps when the training data is sufficient and hurts the accuracy when training data is small in size [18].

On the other hand, we understand that suitable feature space has immense impact on the model learning. If an apt feature set or in our context, appropriate inputs in terms of training data are provided to a suitable deep learning model, the learned model can have better prediction capability. In this paper, we consider Shapley value [19, 20] to estimate the importance of each of the inputs of the model towards the prediction. Shapley values attempt to fairly commensurate a player's contribution in a coalition game. In fact, Shapley value estimation has been applied in diverse disciplines [5, 21]. We incorporate Shapley Value attribution to discard the unnecessary or negatively impacting input data. While augmented learning using adversarial training provides a generic augmentation of the given time series data, the augmented-learned model when gets trained with the suitable input subset using associated Shapley values ensures better learnability. In ShapAAL, data augmentation and input ablation jointly provide the impetus towards learning with better data. ShapAAL can be considered as a push-pull architecture, where augmented learning pushes the model towards getting trained by learning newer (adversarial) examples and Shapley value estimated subset selection pulls the model towards a suitable lower dimension for better learnability and prediction. We introduce the concept of Learn → Unlearn → Re-learn, where the model is initially learned through augmented training; next, Shapley value attribution forces the model to unlearn few detrimental features and subsequently, the model re-learns with the selected subset features using augmented learning. With series of empirical study, we demonstrate the efficacy of our proposed model ShapAAL: Shapley Attributed Augmented Learning and establish the performance superiority over relevant state-of-the-art algorithms.

## Related works

Sensor data-centric classification tasks are mostly likely to undergo the training data scarcity issue owing to its universally acknowledge problem of high expenses and difficulties associated with the generation, collection and the cost with labeling by human experts [22]. Classically, the emphasis was to analyze the time series (given that the senor data is a time series) and to build strong classifiers to solve time series classification tasks [8]. Nearest neighbor based classification with distance function as dynamic time warping distance (1NN-DTW) has been traditionally considered as the classical baseline algorithm for time series classification [23]. COTE or Collective of Transformation-based Ensembles is an ensemble learning algorithm with collection of 35 classifiers [24]. Random Interval Spectral Ensemble (RISE) algorithm builds decision trees with set of Fourier, auto-correlation and partial auto-correlation features and perform ensembling operation [25]. Recently, Time Series Combination of Heterogeneous and Integrated Embeddings Forest (TS-Chief), an tree-based ensemble learning classifier is proposed [26]. In fact, Time Series Forest (TSF) is one of the pioneering works that combine entropy gain with a distance measure to provide evaluation of the split in tree-based ensembling learning [27]. Similarly, Proximity Forest, which is ensembles of highly randomized proximity trees is another ensemble learning algorithm that has been developed for time series classification tasks [28]. Recently, CAnonical Time-series CHaracteristics (Catch22), a feature-engineered time series classifier is proposed that has shown promising results [29]. With deep

learning models showing outstanding performances in computer vision tasks, time series classification also employs strong deep learning architecture like Residual Network (ResNet) [30]. In [31], authors have proposed convolution layer-based residual blocks to develop ResNet-based model, which is considered as a strong baseline for time series classification tasks.

The state-of-the-art techniques as cited above are mostly concerned with the development of a decent time series classification model without the consideration of training data scarcity. It is observed in [22] that time series classification tasks need to emphasis on training data scarcity issue in order to construct a practical analytics system. However, the research direction towards mitigating the learning impairment problem due to training instance insufficiency in time series classification under common machine learning or deep learning framework seems to be an open practical challenge. The typical research attempts are focused towards sophistication of the architecture and detailed extraction of time series representation. Under the constraint of inadequate availability of training set, such attempts may not always be the ideal choice and the diversity of time series applications limit the scalability of such models. In this paper, we propose a novel method ShapAAL, that contains intrinsic capability to demonstrate consistent accurate performance and improves upon the state-of-the art models through the learn, unlearn and re-learn principle of learning with positive impact towards the predictive capability of the model.

In general, machine learning algorithms need to carefully select the supervised features to build a robust model [32]. Optimization method plays an important role in various aspects towards better learned model development under practical constraints [33–39]. For instance, evolutionary processes with consistent equilibrium for high-quality performance and optimization that achieves quicker convergence is proposed in [35]. It is well-known that the search for global optimization in deep learning algorithms often suffer through spurious local optimization issues. In [36], fusion-based meta-heuristic optimization methods are proposed to solve global optimization tasks.

## Materials and methods

### Problem sketch

We focus on time series classification tasks for sensor signal analysis, where typically a time series is represented as an ordered set of real values as: $\boldsymbol{x} = [x_1, x_2, x_3, \ldots, x_T], \boldsymbol{x} \in \mathbb{R}^T$ and $\boldsymbol{x}$ is of length $T$ and $x_1, x_2, x_3, \ldots, x_T$ are the scalar measurements at time intervals $1, 2, 3, \ldots, T$ from a given sensor. For example, an ECG signal $\boldsymbol{x}$ contains continuous time stamp measurements, where the first time stamp measurement is denoted as $x_1$, the second time stamp measurement is denoted as $x_2$ and so on.

Consider a set of $N$ examples that constitute the training dataset $\mathbb{X}_{Train} = [\boldsymbol{x}^{(1)}, \boldsymbol{x}^{(2)}, \ldots, \boldsymbol{x}^{(N)}]$, where each of $\boldsymbol{x}_{(n)}$, $n = 1, 2, \&, N$ is a time series and each of which consists of $T$ number of data samples, i.e. each training instances can be considered as consisting of $T$ number of time stamp measurements from the given sensor. The complete training set consists of corresponding labels: $\mathcal{D}_{Train} = [\mathbb{X}_{Train}, Y_{Train}] = [\{\boldsymbol{x}^{(1)}, y^{(1)}\}, \{\boldsymbol{x}^{(2)}, y^{(2)}\}, \ldots, \{\boldsymbol{x}^{(N)}, y^{(N)}\}]\ y^{(n)} \in [1, \mathbb{C}], \forall n$ are the labels correspond to one of the $\mathbb{C}$ classes. We are particularly concerned to solve the supervised learning tasks for time series classification problems such that a model is constructed from the given input variables or training instances along with its associated labels or ground truths such that model correctly attempts to predict the class that a sensor data belongs to. In supervised learning setting, we find a model or function $h_\theta(.)$, parameterized by $\theta$ that describes the random vector $\mathbf{x}$ associated with label or target $\mathbf{y}$ with joint distribution $p_{data}(\mathbf{x}, \mathbf{y})$. However, we tacitly assume that $[\{\boldsymbol{x}^{(1)}, y^{(1)}\}, \{\boldsymbol{x}^{(2)}, y^{(2)}\}, \ldots, \{\boldsymbol{x}^{(N)}, y^{(N)}\}] \overset{i.i.d.}{\sim} p_{data}$, which means that the

model learning is imposing independently and identically distribution (*i.i.d.*) condition, i.e. the given training examples are drawn independently and identically from $p_{data}$.

In machine learning, the principal aim is to minimize an objective function that penalizes the model $h_\theta(.)$ when it makes mistake, which is denoted by loss function as $L(h_\theta(x), y)$. We consider the expected risk in Eq 1.

$$R(h) = \mathbb{E}_{(\mathbf{x},\mathbf{y}) \sim p_{data}}[L(h_\theta(\boldsymbol{x}), y)] \tag{1}$$

However, we do not have complete idea of $p_{data}(\mathbf{x}, y)$, we simply know the training dataset $\mathcal{D}_{Train} = (\boldsymbol{x}^{(n)}, y^{(n)})$. Hence, we focus on empirical risk minimization (ERM), which is defined in Eq 2.

$$\hat{R}emp(h) = \frac{1}{N} \sum_{n=1}^{N} L(h_\theta(\boldsymbol{x}^{(n)}), y^{(n)})) \tag{2}$$

Considering negative log-likelihood as the loss function under maximum likelihood estimation (MLE) principle, which is a special case of ERM, we the MLE cost function as follows:

$$J(\theta) = \mathbb{E}_{(x,y) \sim \hat{p}_{data}} - log\, p_\theta(\mathbf{y}|\boldsymbol{x})$$

Thus, we need to minimize the cost function $J(\theta)$ to find the parameter $\theta$ from the empirical distribution $\hat{p}_{data}$. The optimization problem is as follows:

$$\theta^* = \arg\min_\theta J(\theta)$$

The practical challenge in time series classification is that $N$ and $T$ are typically much smaller than (or usually as expected by) the corresponding learning methods that solve classical computer vision problems [40]. For example, $N$ can be as low as 50 and often less than 200 and $T$ can be less than 300 for time series classification tasks and on the contrary, classical ImageNet 2012 classification dataset consists of 1.28 million training datasets [9, 10]. However, when $N$ is small (which is profound in time series sensor signal classification tasks), it is not practical to assume the closeness of $\hat{p}_{data}$ and $p_{data}$ and consequently, the learned model tends to get over-fitted to the given training dataset $\mathcal{D}_{Train}$. Hence, the estimation from $J(\theta)$ is poor when the model is directly constructed from the available training dataset $\mathcal{D}_{Train}$.

Let $\eta \in \mathbb{R}$ be the learning rate and the gradient descent function updates the deep neural network model parameter $\theta$ as follows: $\theta \leftarrow \theta - \eta \nabla J(\theta)$. In training data insufficiency problem, the estimation from $J(\theta)$ is incomplete, which in high probability leads the model parameter $\theta$ gets directed towards incomplete or wrongly learned direction from usual gradient descent method. Therefore, we can safely assume that learning degradation is a common problem and approaches that minimize the learning degradation due to training data scarcity needs to be developed in order to achieve better performance from time series classification tasks. Thus, our objective is to find a "good" learned model for diverse set of time series classification tasks (especially where the data is sourced from sensors) in order to minimize the adverse effect of limited training data availability. In a nutshell, the above problem formulation is a generic one that motivates us to build robust time series sensor data classification models, which often suffer due to insufficiency in the training instances.

## Proposed methodology

We propose three stage approach of deep learning model construction, where the first stage learns through augmented learning with additive perturbation of the input samples. Next, the unlearning part identifies the subset of features or the training samples that do not have positive contribution to the model predictive capability through Shapley value computation for each of the input samples. Finally, a new model is re-learned with the subset of samples or with the identified important samples of the training set.

**Model training for augmented learning.** We consider Residual Network or ResNet [30] architecture with controlled perturbation of input space that compensates the lack of training data for time series classification tasks. We consider adversarial perturbation as the set of invariants such that a robust model can be constructed under the practical constraints of training sample scarcity that attempts to minimize the worst-case classification error due to the data perturbation by the adversary [15]. The adversary, in turn augments the training space as an automated (machine generated) labeler, replacing the human labeler. Hence, we not only gain in the enrichment of the training process, but also avoid the expensive process of collection and labeling of time series examples. The adversarial perturbation is to force the classifier to learn hidden representations of unseen neighbor feature in order to estimate the true distribution $p_{data}$. Let $J_{adv}(\theta_{adv}, x, y)$ be the cost (associated with the adversarial loss $L_{adv}$ for training the network (in our case, we primarily consider the neural network as ResNet [30]) to derive the model parameter $\theta_{adv}$. ResNet has shown tremendous success in different classification tasks. It aims to tackle the learnability issue of deep neural networks by minimizing the exploring and vanishing gradient problems through norm preservation of error gradient [41]. ResNet transforms the traditional representation learning to learn $\mathcal{H}(x) = \mathcal{F}(x) + x$ at each layer [30] as depicted in Fig 2, where one typical Residual Block (RB) is shown. The main advantage of $\mathcal{H}(x)$ is to ensure that the information in $x$ flows throughout the network [41]. In ResNet, the original mapping is recast into $\mathcal{F}(x) + x$ [30] and it is hypothesized that optimization of the residual mapping becomes easier [30]. In fact, the identity or shortcut connection does the desirable effect of norm preservation as error gradient [41] as shown in Fig 1. In ShapAAL, we transform $x \rightarrow x + \delta$ for perturbed identity and it learns through

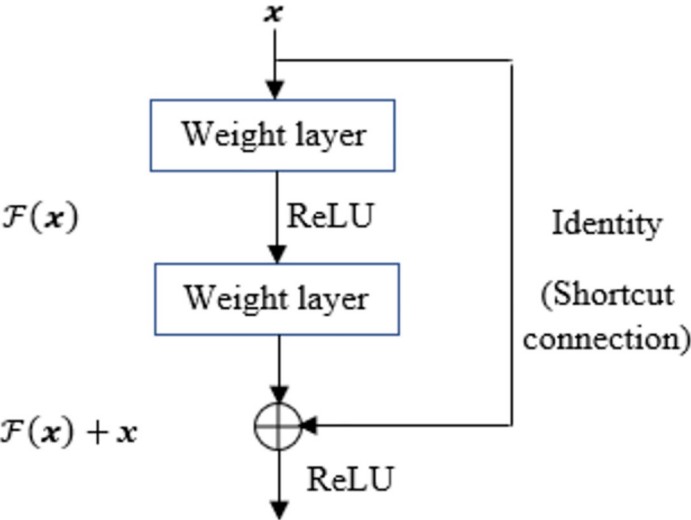

**Fig 1. A residual block (RB) in ResNet ([30]).**

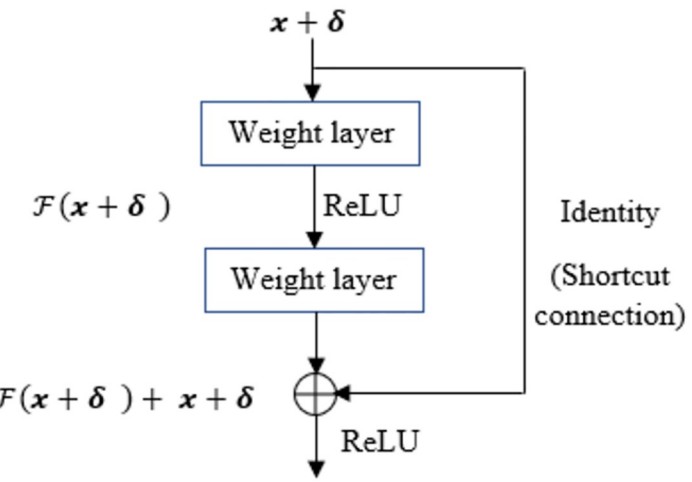

**Fig 2. A residual block (RB) in ShapAAL with perturbed input.**

$\mathcal{H}(x + \delta) = \mathcal{F}(x) + x + \delta$. Therefore, the transformation of $x \to x + \delta$ into the identity connection augments the learnability through perturbation-induced shortcut connection. The identity connection becomes perturbation connection as shown in Fig 2.

We design the ShapAAL model with ResNet architecture using restrained learning principle [42]. It consists of variable number of residual blocks (RB) between 10 and 2 and the residual block depth (i.e. the number of total RBs in the model) depends on the training data. We estimate the network depth (measured in term of number of RBs) using restrained learning principle that analyzes the training dataset distribution to adjust the network depth. We depict the typical RB of ShapAAL in Fig 3.

We convert the 1D data to 2D through reshape operation such that the features of 2D convolutions can be utilized. The batch size is variable, depending upon the number of training instances. The batch size is calculated as: $min\left(ceil\left(\frac{number\ of\ training\ instances}{10}\right),\ 16\right)$. When the training examples are small in number ($\leq 10$), batch size of 2 is considered. We consider fixed learning rate $10^{-3}$, which is default in Keras. We $z$-normalized the training data as: $\frac{\mathbf{X}_{Train} - mean(\mathbf{X}_{Train})}{standard\ deviation\ (\mathbf{X}_{Train})}$ as well as the test data as: $\frac{\mathbf{X}_{test} - mean(\mathbf{X}_{Train})}{standard\ deviation\ (\mathbf{X}_{Train})}$. We first calculate $x + \delta$ and after that $z$-normalization is performed. Please note that the statistical estimation of $z$-normalization operation for test data is made from the provided train data as the statistics of test data is unknown. After the final residual block, Global Average Pooling is used. We use softmax function in the output layer for the classification task and cross-entropy as the loss function. The input data for identity connection is $x + \delta$. Depending upon the estimation of residual block depth from restrained learning algorithm [42], the number of RBs are constructed. Let, the number of RBs be $\chi$, where $10 \leq \chi \leq 2$. For different $\mathbf{X}_{Train}$, the value of $\chi$ might be different owing to the differences of the underlying training distribution and accordingly, the model with $\chi$ number of RBs is constructed. We illustrate ShapAAL architecture in Fig 4.

The expected risk of ShapAAL under augmented learning is defined in Eq 3.

$$R_{aug}(h) = \mathbb{E}_{(\mathbf{x},\mathbf{y}) \sim \hat{p}_{data}}\left[\max_{\delta \in \Delta} L(h_\theta(x + \delta), y)\right] \tag{3}$$

where $\Delta$ represents the set of adversarial perturbations in $\delta$ to induce mis-classification. The input is perturbed with noise $\delta$ such that the network gets the opportunity to learn training

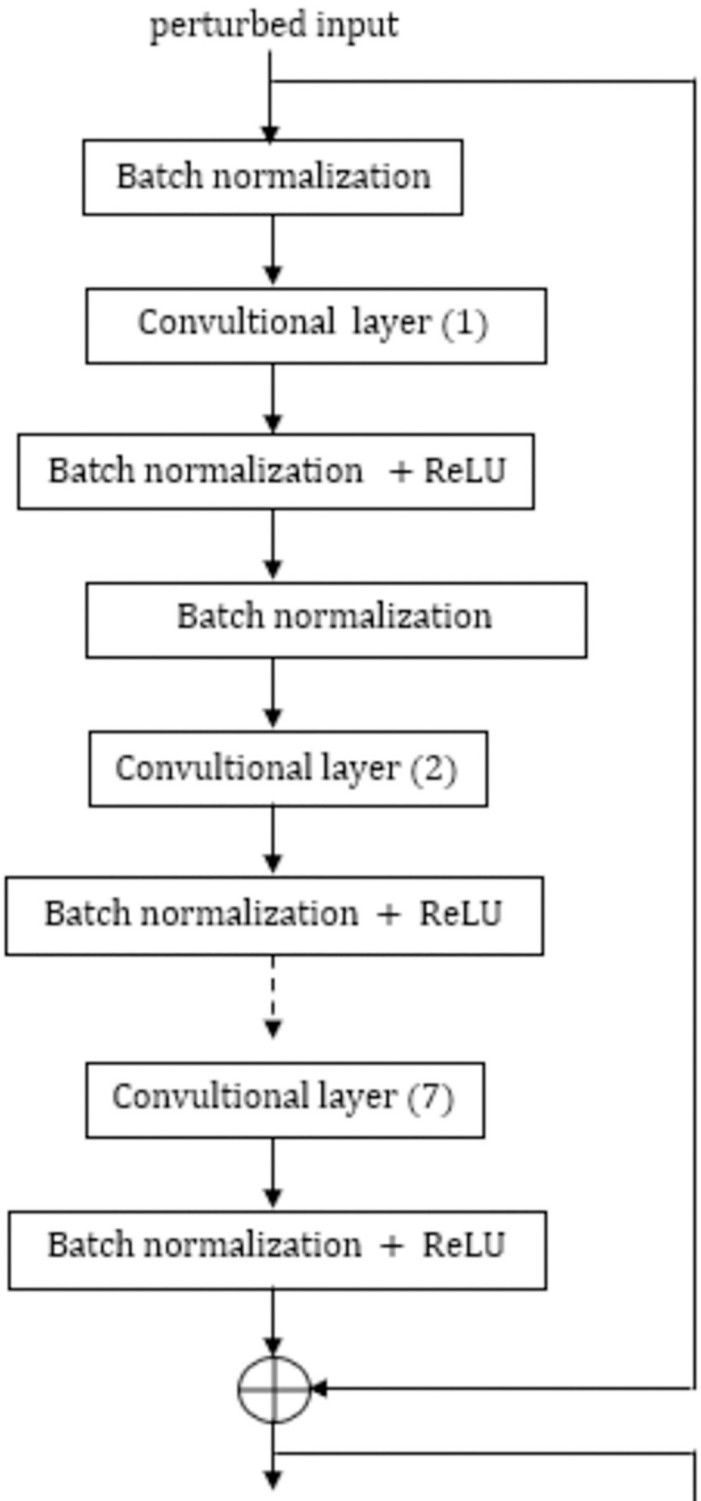

**Fig 3. Single residual block in ShapAAL architecture.**

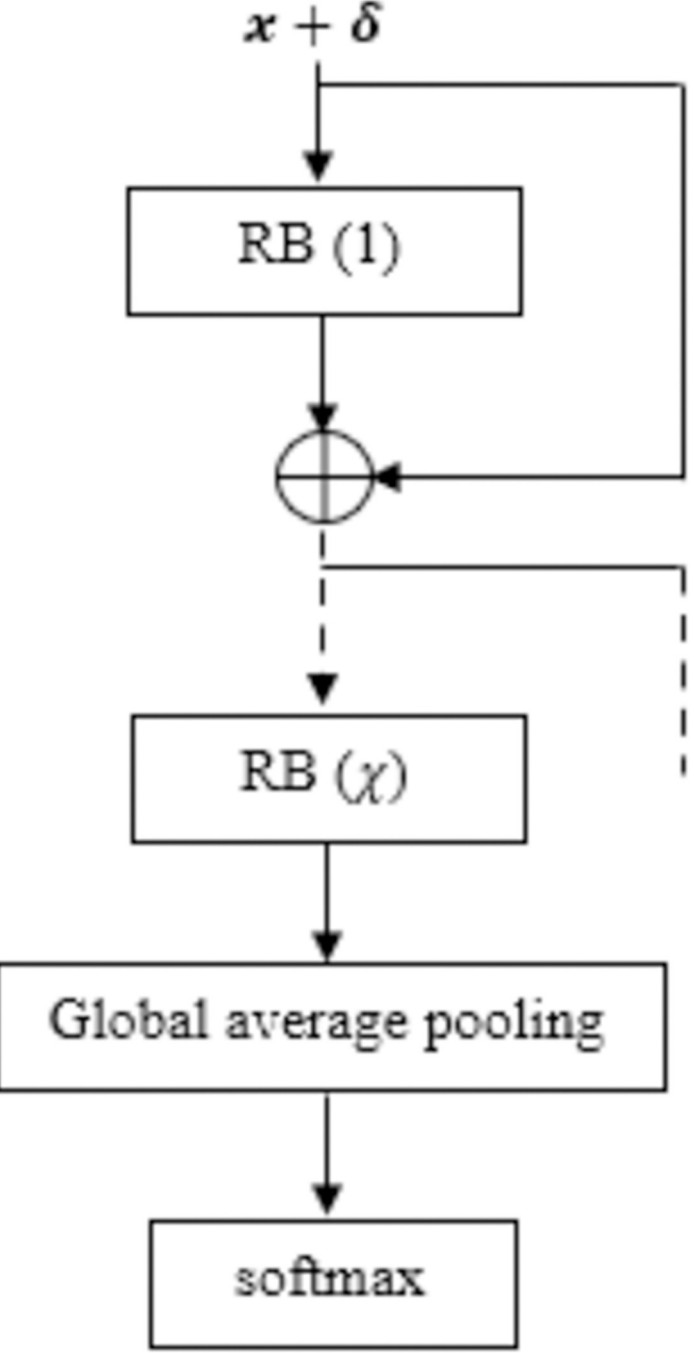

**Fig 4. ShapAAL architecture.**

examples outside the given training set while the unperturbed shortcut connection makes the gradient to avoid being trapped into the spurious local optimum [43]. Hence, we hypothesize that the network learns well with the identity propagation of a ResNet model through shortcut connection that guides the algorithm to move easily towards global optimum [43] and the perturbed input space forces the model to learn unseen examples through augmented learning

gain. The perturbation needs to be controlled and introduction of controlled additive perturbation compels the learning to be more generic and it learns examples beyond the given training data. We perturb the input data by adding small amount of Gaussian noise $\delta$ and the parameters (mean and standard deviation) of the Gaussian noise $\delta$ are derived from $\mathcal{X}_{Train}$, i.e. $\delta$ is sampled from $\mathcal{N}(\mu, \sigma^2)$, where, $\mu$ being the mean of $\mathcal{X}_{Train}$ and $\sigma^2$ being the variance of $\mathcal{X}_{Train}$. In order to maintain a reasonably high signal-to-noise ratio between the perturbed data and original data, a scaling factor $\alpha$ is introduced and $\delta$ is sampled from $\alpha \times \mathcal{N}(\mu, \sigma^2)$. In fact, we change the view of a training set to a learner system such that the leaner's robustness is examined and eventually we expect a stronger model with higher generalization gain and lower over-fitting error. We have considered $\alpha = 0.020$ throughout the experimental process. The controlled additive perturbation enforces the ResNet model to learn the less confident solutions to lower the generalization loss. Hence, the learnability of the model improves when it faces newer types of challenges (as expected in the field when unseen test data are encountered).

We attempt to generate the augmented learning model to minimize the generalization loss by introducing perturbation into the learning space. Further, we incorporate restrained learning for adjusting the depth of the network (more precisely, the number of residual blocks), which is training data distribution-aware [42]. For a given training data $\mathcal{D}_{Train} = [\mathbb{X}_{Train}, Y_{Train}]$, we estimate the network depth through restrained learning approach of elastic depth estimation [42]. Elastic depth minimizes the negative impact of data perturbation. When the perturbed training data results in redundancy, the network depth shrinks and vice-versa. The restrained learning which dynamically configures the network depth acts as a regularizer to restrict the learning when the data redundancy due perturbation process is high. Let us denote the adversarially trained augmented model with adversarial risk $R_{aug}(h)$ minimization as $M_{aug}$.

**Subset selection from input samples for model re-learning.**   Augmented training has the advantage of better learning due to perturbation in the learning process, but such learning may not always do the good for model predictability. Using adversarial training for data augmentation requires to know the worst-case $\delta$ that augments the training data "most beneficial" way with "highest confusion creation" to the training data [44]. However, such search is computationally (extremely) expensive. We propose that an apt process of important feature selection or sampling the input data ensures the required better learning for the model. Hence, we identify $\mathcal{D}_{Trainsub} = [\mathbb{X}_{Trainsub}, Y_{Trainsub}]$ which is a subset of $\mathcal{D}_{Train}$, i.e. $\mathcal{D}_{Trainsub} \subseteq \mathcal{D}_{Train}$ such that the samples with positive impact on the model predictability are chosen.

Our objective is to estimate the importance of a feature or training sample $\boldsymbol{x}^{(n)} \subset \mathbb{X}_{Train}$ such that the worth of $\boldsymbol{x}^{(n)}$ is significant to consider it as an important and positively contributing sample. We use Shapley value [19, 20], a fundamental concept in transferable utility cooperative game theory [45] to quantify the attribution of $\boldsymbol{x}^{(n)}$ in the prediction capability of the constructed model. Let $N$ be a finite set of training samples (in cooperative game theory context, we call the training samples as players) or player [46, 47].

1. **(Definition I)** (Transferable utility game). We define a game that maps $v:2^N \rightarrow \mathbb{R}$ such that $v(\emptyset) = 0$. We interpret $v(\psi)$ where $\psi$ in $2^N$, as the estimated value of coalition $\psi$. The value function $v(\psi)$ intends to identify the collective payoff a player's or a set of players' gain when they cooperate and the model $M$ is trained with $n^{th}$ sample on all possible subset $\psi \subseteq 2^N$.

2. **(Definition II)** (Marginal contribution). We define the marginal contribution $\Delta_v(n, \psi)$ of player $n$ with respect to the coalition $\psi$ as: $\Delta_v(n, \psi) = v(\psi \cup n) - v(\psi)$.

With $\Lambda$ being denoted as the integer permutations up to $N$ and $\lambda \in \Lambda$ and we represent the predecessor set of players preceding $n^{th}$ player in $\lambda$ as: $\psi_{n,\lambda} = \{m: \lambda(m) < \lambda(n)\}$. With this

definition, Shapley value $\varphi_n$ of $n^{th}$ player is formulated as the weighted average of the marginal contribution of it to all other possible subset of players in the game. Accordingly, Shapley value $\varphi_v(n)$ of $n^{th}$ player with the function $v$ is:

$$\phi_v(n) = \frac{1}{N!}\sum_{\lambda \in \Lambda}\Delta_v(n, \psi_{n,\lambda}).$$

From the permutation logic, we can compute the Shapley value $\varphi_v(n)$ of $n^{th}$ training sample as:

$$\phi_v(n) = \frac{1}{N!}\sum_{\psi \subseteq \{1,2,3..,N\}}|\psi|!(N - |\psi| - 1)!\Delta_v(n, \psi).$$

The above equation needs to be solved to get the estimation of Shapley value for each of the training examples in $N$, but that process is computationally expensive. In this paper, we consider the high-speed approximation of $\varphi_v(n)$ using DeepLIFT algorithm [48] with DeepExplainer implementation (https://github.com/slundberg/shap).

From the computed Shapley values $\varphi_v(n), \forall n \in N$ for each of the training samples in $\mathcal{D}_{Train} = [\mathbb{X}_{Train}, Y_{Train}]$, we discard the negative valued ones, i.e. the training samples which contain negative magnitude in their Shapley value are removed and new training set with $N^{effective} \leq N$ number of training examples are formed and the expected risk of ShapAAL is depicted in Eq 4.

$$R_{ShapAAL}(h) = \mathbb{E}_{(\mathbf{x},\mathbf{y})\sim\hat{p}_{data}}[\max_{\delta \in \Delta} L(h_\theta(\mathbf{x}_s + \delta), y_s)] \tag{4}$$

where, $\{\mathbf{x}_s, y_s\}$ belong to the Shapley value attributed dataset. A set of axioms namely "efficiency" and "null player" are the prime motivations to claim that the context of Shapley value for finding out the right subset [46, 47].

1. **(Axiom I)** (Fairness). The worth of a complete model $v(N)$ in a transferable utility game is a lossless distribution among the given features: $\Sigma_{n \in N}\varphi(n) = v(N)$.

2. **(Axiom II)** (Null player). If a feature $n$ contributes nothing in a transferable utility game $v$, its Shapley value is zero. $[(\forall \psi)v(n \cup \{n\}) = v(\{n\})] \Rightarrow \phi(n) = 0$.

**Axiom I** and **Axiom II** help us to develop the subset selection algorithm. Let us denote the newly formed training set with $N^{effective}$ number of training samples as $\mathcal{D}_{Train\_effective} \subseteq \mathcal{D}_{Train}$. The unlearning part gets completed with newly formed training set $\mathcal{D}_{Train\_effective}$. The model previously learned with $\mathcal{D}_{Train}$ with adversarial risk $R_{aug}(h)$ (Eq 3) minimization as $M_{aug}$, subsequently, re-learns $\mathcal{D}_{Train\_effective}$ by minimizing $\textit{R}_{ShapAAL}(h)$ (Eq 4) to construct ShapAAL model $M_{aug}^{ShapAAL}$ through unlearning the negatively impacting dataset. When training dataset is large, the negative contribution of few data may not have some impact, but in case of smaller number of training datasets, the negatively contributing ones can have higher impact on the learning of the model. Classically, the model learning flow is: training data → model training → classification by the trained model. With data augmentation training the flow is: training data → augmentation → augmented model training → classification by the augmented trained model. With Shapley value-vbased feature attribution the training flow is: Training data → Subset selection from the knowledge of Shapley values of each of the input data → Shapley-attributed model training → Classification by the Shapley-attributed model. We propose the model training algorithm ShapAAL that takes advantage of augmented training for training space augmentation as well as subset selection through Shapley value-attribution as defined in Eq 4. The proposed model training

flow is: training data $\rightarrow$ augmentation $\rightarrow$ augmented model training $\rightarrow$ Subset selection from the augmented set wit the knowledge of Shapley values $\rightarrow$ Shapley-attributed augmented model training $\rightarrow$ classification by the Shapley-attributed augmented model. We depict the ShapAAL Algorithm 1 below.

1. Construct the model $M_{aug}$ with adversarial risk of augmented learning $R_{aug}(h)$ from Eq 3 through risk minimization from the given training dataset $\mathcal{D}_{Train} = [\mathbb{X}_{Train,}Y_{Train}]$, with $N$ number of training examples according to the ShapAAL architecture in Figs 2–4, where input is perturbed with noise $\delta$ to provide the network with the capability to learn outside the given training example set.
   Learning part

2. With the model $M_{aug}$ as reference, for each of the training instances $n \in N$, $\varphi(n)$ Shapley values are computed from DeepLIFT algorithm [48].

3. We find those $n$ where $\varphi(n) \leq 0$, **(Axiom II)** which create the set of $N'$ number of examples, $N' \leq N$.

4. Discard those $N'$ number of training samples and rest $N^{effective}$ training samples create new training dataset $\mathcal{D}_{Train\_effective}$, where, $N^{effective} = N\text{-}N'$.
   Unlearning part

5. ShapAAL model $M_{aug}^{ShapAAL}$ is generated by training with $\mathcal{D}_{Train\_effective}$ containing $\{\boldsymbol{x}_s, y_s\}$ according to the ShapAAL architecture in Figs 2–4, where Shapley attributed inputs $\boldsymbol{x}_s$ are additively perturbed with noise $\delta$ to construct ShapAAL model $M_{aug}^{ShapAAL}$ by minimizing $R_{ShapAAL}(h)$ from Eq 4.
   Re-learning part

In summary, we define a transferable utility game for the selecting useful training data, which are made inputs to the learning algorithm. The attribution of each of the inputs into the model predictability is estimated through Shapley value computation. The non-contributing inputs defined according to **Axiom II** are discarded as the not worthy inputs and the remaining inputs are used for re-learning the model. The subset finding operation is performed over the perturbed set with the assumption that the perturbed input space provides augmentation when the training data is insufficient.

## Results

We conduct series of empirical studies to investigate the performance efficacy of ShapAAL in time series sensor data classification tasks particularly when the training data sample size is small.

### Data description

Currently, UCR [8] is one of the most recognized time series classification benchmark archives [49]. We find number of time series sensor datasets along with three important ECG datasets which fulfill our criteria of being limited in number of training instances ($\leq$200). The datasets are sourced from sensing devices. These datasets are diverse in different characteristics like sensor types, number of training examples, length of the data etc. as depicted in Table 1. Each of the time series datasets in UCR has fixed and exclusive training and testing splits. The test data is completely hidden. In this work, we have generated the learning model using the training datasets and the trained model is tested on the provided testing datasets and the 'test

**Table 1. Experimental dataset description.**

| Dataset | Training size | Data length | Sensor type | Application type |
|---|---|---|---|---|
| ChinaTown | 20 | 24 | IR Sensor | Pedestrian counting |
| Coffee | 28 | 286 | Food Spectrometer | Detection of two different coffee types for food safety and quality assurance |
| ECG200 | 100 | 96 | Electrocardiogram (ECG) | Myocardial infarction detection |
| ECGFiveDays | 23 | 136 | Electrocardiogram (ECG) | Change detection |
| FreezerRegularTrain | 150 | 301 | Smart energy meter | Engery Efficiency in domestic appliances |
| FreezerSmallTrain | 28 | 301 | Smart energy meter | Engery Efficiency of freezers |
| ItalyPowerDemand | 67 | 24 | Smart energy meter | Power demand identification |
| MoteStrain | 20 | 84 | Humidity sensor | Weather feature |
| PowerCons | 180 | 144 | Smart energy meter | Household energy consumption pattern identification |
| SonyAIBO1 | 20 | 70 | Accelerometer | Surface type identification |
| SonyAIBO2 | 27 | 65 | Accelerometer | Surface type identification |
| TwoLeadECG | 23 | 82 | Electrocardiogram (ECG) | Change detection |

accuracy' (as per the convention in the UCR time series archive benchmark [8, 49]) is considered as the classification inference performance measure.

## Development environment

ShapAAL is implemented in Keras 2.1.2 on Python 3.5.4 on Tensorflow 1.4.0 library. The hardware environment for training the model consists of 64-bit x86 architecture 16 cores Intel Xeon CPU E5–2623 v4 with 2.60GHz clock speed with two Nvidia GeForce GTX 1080 GPUs, which are powered by Pascal architecture and each of the GPUs has 10 GB memory. We have used DeepExplainer implementation of DeepLIFT algorithm [48]. In DeepExplainer (https://github.com/slundberg/shap), a distribution of background samples is used instead of a single reference point in DeepLIFT. In order to minimize the impact of non-reproducibility (https://glaringlee.github.io/notes/randomness.html) with run-to-run variability due to nondeterminism in neural networks [50, 51], we have considered at least 50 different random seeds for each of the experimental datasets and the reported empirical results are the highest occurring (mode) of the obtained test accuracies.

## Empirical investigation

We perform number of empirical investigations including ablation study, comparative study with relevant baselines and state-of-the-art algorithms to illustrate the practical utility of ShapAAL when performing diverse set of real-world sensor time series classification tasks including that of critical prediction task of Myocardial Infarction condition detection from ECG sensor. We show in Figs 5–9 that the Shapley value responses of the training instances in different datasets. It is clearly observed that few of the training instances are in fact negatively impacting towards the model prediction and it is evident in practice that each training samples are not essentially contributing positively towards model's prediction.

Subsequently, in Fig 10, we depict the distribution of subset selection from Shapley attribution in ShapAAL, where in some cases, more than 30% of training samples are rendered unimportant and subsequently discarded in the process of learning.

Next, we conduct ablation study to understand the efficacy of the proposed model. An ablation study in general, investigates the performance of a machine learning system by removing few components in order to evaluate the impact of those components in the complete system. Similarly, ShapAAL model construction consists of four components that include the base

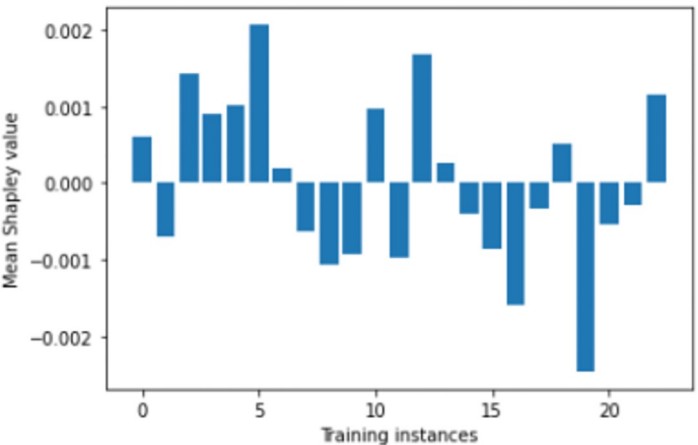

**Fig 5. Estimation of input attribution for "ECGFivedays" dataset.**

model (ResNet), Shapley value attribution over the base model, data augmented training on the base model and data augmented training with Shapley attributed feature selection on the base model. We denote $M$ as the base model that is trained with each of the training data, $M^{Shapley}$ as the model that is trained with the training data after discarding the negatively contributing Shapley valued features, $M_{aug}$ is the model that is adversarially trained over over entire augmented training data. $M^{ShapAAL}$ or $M_{aug}^{Shapley}$ is the adversarially trained with the augmented training data with discarding the negatively contributing Shapley valued ones following the deep architecture in Fig 4 as depicted in Figs 11–14. In Table 2, we depict the "test accuracy" performances of $M$, $M_{aug}$, $M^{Shapley}$ and $M_{aug}^{ShapAAL}$ over the experimental datasets. The ablation study unambiguously indicates that our proposed model $M^{ShapAAL}$ is the superior one. In fact, the trend is also clear that both augmented training and Shapley attributed re-learning have significant positive impact on the learnability of the model, which reflects in the consistent superlative performance of $M^{ShapAAL}$ w.r.t the others. Conceptually, the ShapAAL model is evolved from the base model $M$, which learns from $R(h) = \mathbb{E}_{(\mathbf{x},\mathbf{y}) \sim p_{data}}[L(h_\theta(\boldsymbol{x}), y)]$.

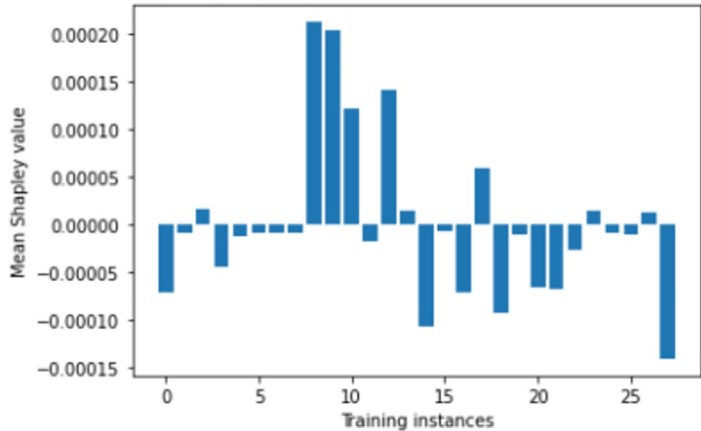

**Fig 6. Estimation of input attribution for "FreezerSmallTrain" dataset.**

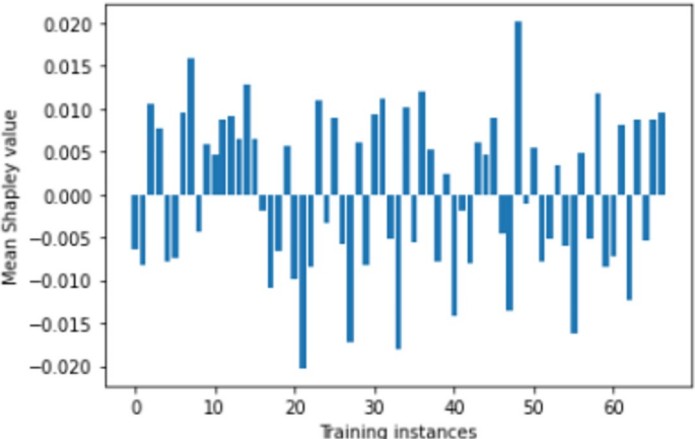

**Fig 7. Estimation of input attribution for "ItalyPowerDemand" dataset.**

$M_{aug}$ model, derived from $M$ directly helps the base model $M$ to get trained over unseen training examples due to additive perturbation with benefit of addressing training data scarcity problem by learning from $R_{aug}(h) = \mathbb{E}_{(\mathbf{x},\mathbf{y}) \sim \hat{p} \, data}[\max_{\delta \in \Delta} L(h_\theta(\boldsymbol{x} + \delta), y)]$. On the other hand, model $M^{Shapley}$ gets trained over a subset of the seen training examples according to Shapley-value attribution that discards the non-important input features. Our proposed model considers the strengths of both $M_{aug}$ and $M^{Shapley}$ to construct an unique deep learning algorithm that renders data augmentation as well as input feature reduction (i.e. getting advantages from adversarial training and apt feature selection) to allow the ResNet base model $M$ to appropriately learn over augmented yet selected input set. Hence, we establish with the empirical support that less number of input features (Refer Fig 10) when properly selected can provide better test accuracy. Under training data size constraint scenario, the push-pull architecture of ShapAAL as a coalition game with Shapley attributed push towards lower dimension and concurrently pulling or augmenting the learning capability of the model over unseen data indeed demonstrates significantly improved performance.

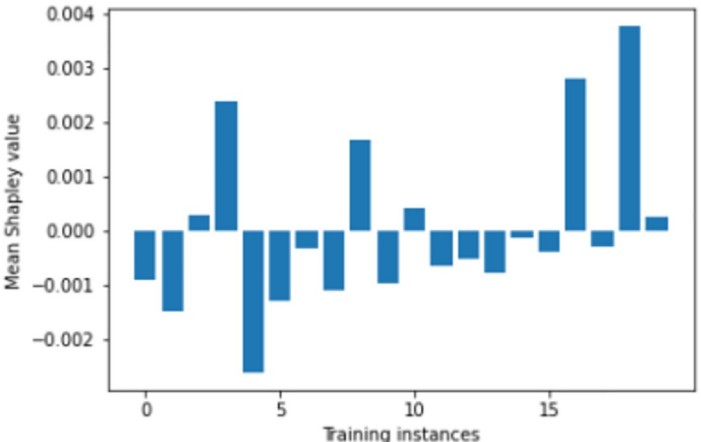

**Fig 8. Estimation of input attribution for "MoteStrain" dataset.**

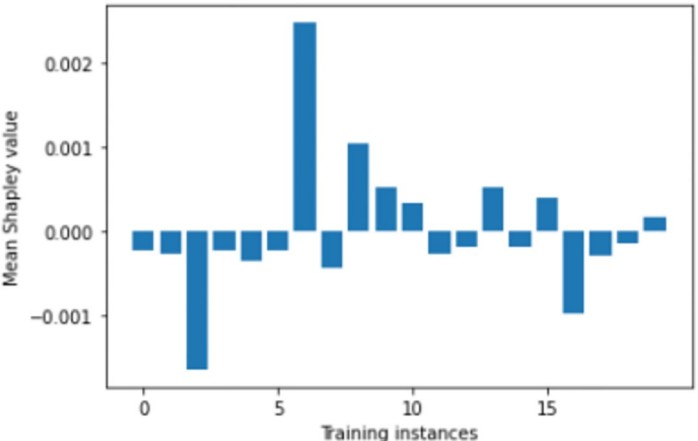

**Fig 9. Estimation of input attribution for "SonyAIBOSurface1" dataset.**

Given that generic time series classification is well-studied [8], we do an exhaustive comparative study with the baseline algorithms like 1NN-DTW-based model [23] as well as state-of-the-art methods including RISE [25], COTE [24], TS-Chief [26], Time Series Forest (TSF) [27], Proximity Forest (PF) [28], Catch22 [29], and time series ResNet [31]. In Table 3, the comparative study of test accuracies of relevant state-of-the-art algorithms are shown and we observe that ShapAAL consistently outperforms the state-of-the-art algorithms.

Another classical performance merit is the "outperforming" the benchmark. In recent years, number of time series classification algorithms have been proposed in literature, which might not have been updated in the UCR archive repository. However, we can consider the available benchmark or the best results in the UCR repository of the respective datasets as the

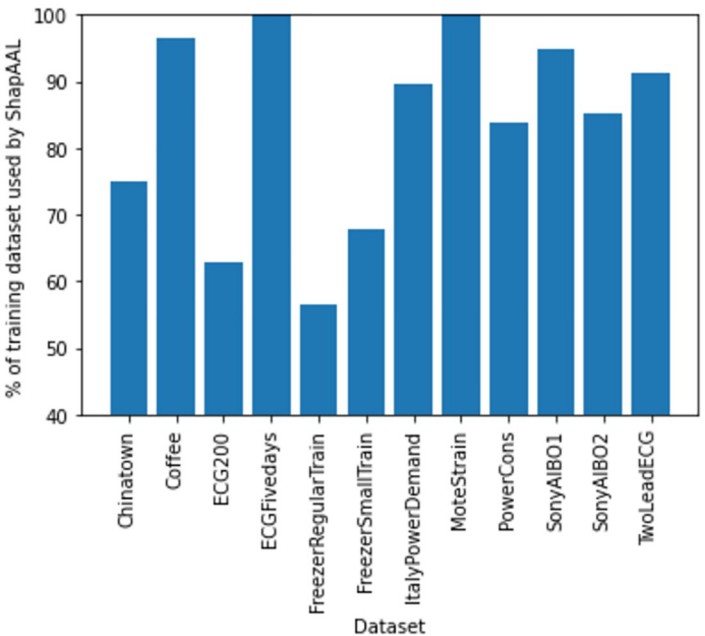

**Fig 10. Selection of subset by ShapAAL algorithm for different datasets.**

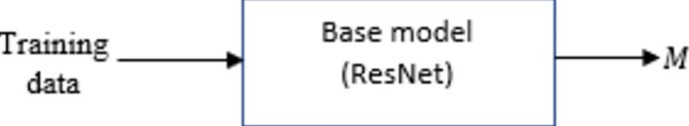

**Fig 11. Constructing the base model $M$.**

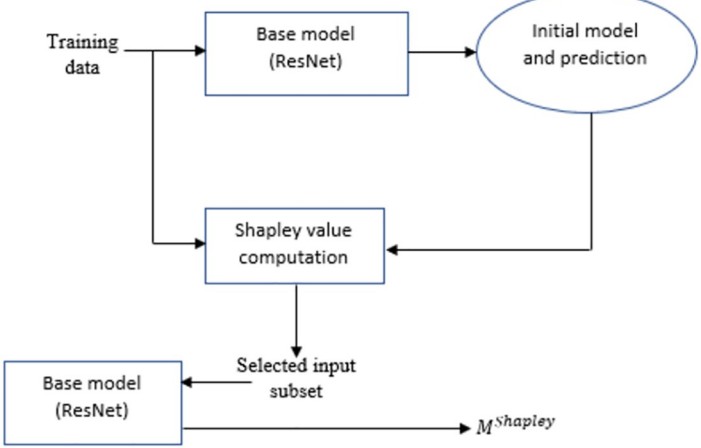

**Fig 12. Constructing the Shapley-value ablated model $M^{Shapley}$.**

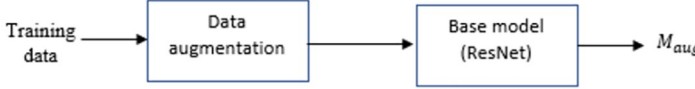

**Fig 13. Constructing the augmented learning model $M_{aug}$.**

"reported benchmark". In Fig 15, we depict the differential test accuracy gain of the algorithms (which has reported results available in public domain) including ShapAAL model w.r.t the reported best results and it is computed as $\frac{test\ accuracy\ of\ the\ algorithm\ -\ reported\ benchmark\ test\ accuracy}{reported\ benchmark\ test\ accuracy}$ with the aim of being the test accuracy result to be positive, indicating that the concerned algorithm has

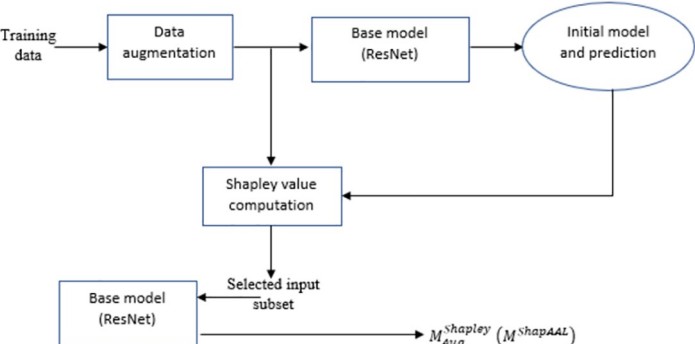

**Fig 14. Constructing the proposed model $M^{ShapAAL}$.**

**Table 2. Ablation study through test accuracies of ShapAAL model ($M^{ShapAAL}$) with $M$, $M^{Shapley}$, $M_{aug}$.**

| Algorithm | $M$ | $M^{Shapley}$ | $M_{aug}$ | $M^{ShapAAL}$ |
|---|---|---|---|---|
| ChinaTown | 0.890 | 0.901 | 0.9211 | 0.9722 |
| Coffee | 0.976 | 1.00 | 0.998 | 1.00 |
| ECG200 | 0.83 | 0.86 | 0.87 | 0.92 |
| ECGFiveDays | 0.989 | 1.00 | 1.00 | 1.00 |
| FreezerRegularTrain | 0.9865 | 0.9901 | 0.9933 | 0.9984 |
| FreezerSmallTrain | 0.8640 | 0.8640 | 0.8613 | 0.9309 |
| ItalyPowerDemand | 0.8910 | 0.8901 | 0.9356 | 0.9704 |
| MoteStrain | 0.8101 | 0.8233 | 0.9087 | 0.9084 |
| PowerCons | 0.8576 | 0.8571 | 0.9083 | 0.9633 |
| SonyAIBO1 | 0.8121 | 0.8439 | 0.8907 | 0.9682 |
| SonyAIBO2 | 0.9355 | 0.9451 | 0.9406 | 0.9461 |
| TwoLeadECG | 0.8860 | 0.9006 | 0.9304 | 0.9994 |

outperformed the currently reported benchmark result. We observe that proposed ShapAAL steadily outperforms the reported benchmark results in comparison with the relevant benchmark algorithms.

Mean Per-Class Error (MPCE) ([31])is another useful metric to evaluate the classification performance of the model as: the expected error rate for a single class across each of the test data. For $Y$ number of test data with class $c_v$ and corresponding error rate $err_v$, we compute MPCE as: $\frac{1}{Y}\sum \frac{err_v}{c_v}$.

MPCE seems to a robust as an evaluator of model performance for different datasets of the classes [31]). Below in Table 4, we demonstrate the MPCE results for the ablation study. In MPCE, our aim is to have a lower value, approaching zero.

Another unique feature of the current work is its response to higher number of test instances when it gets trained with smaller number of training examples. We can quantify the learning gain of ShapAAL at the time of testing as: $\frac{test\ accuracy_{ShapAAL} - test\ accuracy_{Base}}{test\ accuracy_{Base}}$ and also define training insufficiency factor as: $\frac{Number\ of\ training\ examples}{Number\ of\ testing\ instances}$. In Fig 16, we demonstrate the comparative

**Table 3. Comparative study of test accuracies of ShapAAL model with baseline and state-of-the-art algorithms INN-DTW ([23]), COTE ([24]), TS-Chief ([26]), ResNet ([31]), PF ([28]), RISE ([25]), TSF ([27]), Catch22 ([29]).**

| Algorithm | INN-DTW([23]) | COTE([24]) | TS-Chief ([26]) | ResNet([31]) | PF([28]) | RISE([25]) | TSF([27]) | Catch22([29]) | ShapAAL |
|---|---|---|---|---|---|---|---|---|---|
| ChinaTown | * | * | 0.9618 | 0.9701 | 0.94801 | 0.8885 | 0.9529 | 0.9344 | 0.9722 |
| Coffee | 0.821 | 1.00 | 0.9904 | 0.9964 | 0.9916 | 0.9845 | 0.9869 | 0.9797 | 1.00 |
| ECG200 | 0.88 | 0.88 | 0.855 | 0.8836 | 0.873 | 0.851 | 0.86 | 0.7886 | .92 |
| ECGFiveDays | 0.7967 | 1.00 | 0.9988 | 0.9510 | 0.8828 | 0.97286 | 0.9519 | 0.8158 | 1.00 |
| FreezerRegularTrain | * | * | 0.9984 | 0.9967 | 0.9423 | 0.9522 | 0.9970 | 0.9981 | 0.9984 |
| FreezerSmallTrain | * | * | 0.9954 | 0.9494 | 0.8233 | 0.8787 | 0.9614 | 0.9597 | 0.9309 |
| ItalyPowerDemand | 0.9553 | 0.9611 | 0.9624 | 0.9571 | 0.9560 | 0.9445 | 0.9594 | 0.8774 | 0.9704 |
| MoteStrain | 0.8658 | 0.9369 | 0.9301 | 0.9031 | 0.9149 | 0.8780 | 0.8554 | 0.8484 | 0.9084 |
| PowerCons | * | * | 0.9794 | 0.8861 | 0.9874 | 0.9579 | 0.9931 | 0.8862 | 0.9633 |
| SonyAIBO1 | 0.6955 | 0.8453 | 0.8897 | 0.9603 | 0.9201 | 0.8669 | 0.8637 | 0.8833 | 0.9682 |
| SonyAIBO2 | 0.8594 | 0.9517 | 0.9010 | 0.9688 | 0.8990 | 0.9124 | 0.8743 | 0.9023 | 0.9461 |
| TwoLeadECG | 0.86 | 0.993 | 0.9900 | 0.9994 | 0.9817 | 0.9107 | 0.8706 | 0.8539 | 0.9994 |

* marked results are not available.

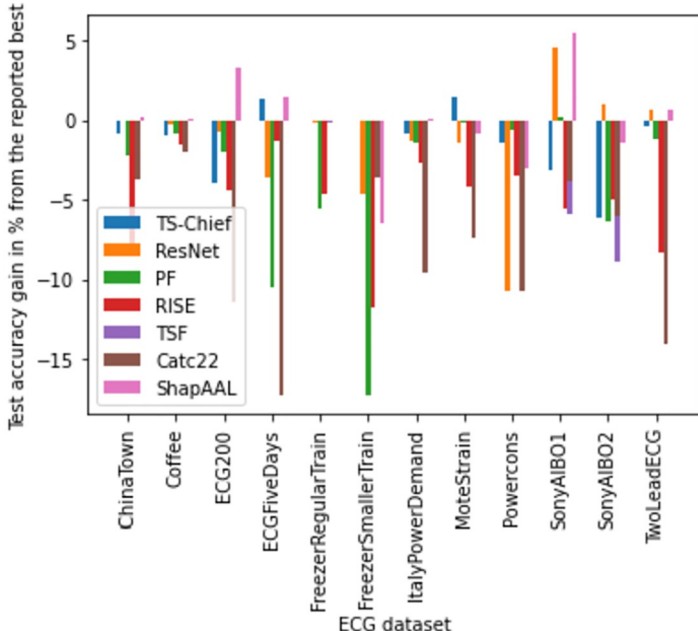

**Fig 15. Differential test accuracy gain of different algorithms and proposed ShapAAL from the current reported (best) benchmark results.**

study of learning gain of ShapAAL on testing data over base model and the insufficiency in the training. We observe that the learning gain of ShapAAL is mostly $\geq 1$, while training insufficiency factor $\leq 1$. Hence, we further establish our claim that ShapAAL model is the apt choice under practical constraint of training data limitation in solving the time series classification tasks.

The significance of ShapAAL as a time series sensor data classification model is well-established both from ablation study (Table 2) and comparative study with current state-of-the-art algorithms (Table 3, Fig 15). ShapAAL not only improves upon through joint augmented training and Shapley value based feature attribution, but also it creates new benchmark in time series sensor signal classification tasks. With the support of the above empirical study, we

**Table 4. Ablation study through MPCE of ShapAAL model ($M^{ShapAAL}$) with $M$, $M^{Shapley}$, $M_{aug}$.**

| Algorithm | $M$ | $M^{Shapley}$ | $M_{aug}$ | $M^{ShapAAL}$ |
|---|---|---|---|---|
| ChinaTown | 0.1115 | 0.1002 | 0.0751 | 0.0281 |
| Coffee | 0.0409 | 0.0 | 0.011 | 0.0 |
| ECG200 | 0.1785 | 0.1454 | 0.1417 | 0.0846 |
| ECGFiveDays | 0.0174 | 0.00 | 0.0 | 0.00 |
| FreezerRegularTrain | 0.0119 | 0.0085 | 0.0079 | 0.0021 |
| FreezerSmallTrain | 0.1587 | 0.1587 | 0.1532 | 0.0879 |
| ItalyPowerDemand | 0.1089 | 0.1082 | 0.0649 | 0.0305 |
| MoteStrain | 0.1981 | 0.1772 | 0.1045 | 0.1049 |
| PowerCons | 0.1459 | 0.1457 | 0.1006 | 0.0457 |
| SonyAIBO1 | 0.1870 | 0.1640 | 0.0995 | 0.0401 |
| SonyAIBO2 | 0.0615 | 0.0604 | 0.0588 | 0.0583 |
| TwoLeadECG | 0.1239 | 0.0902 | 0.0689 | 0.0012 |

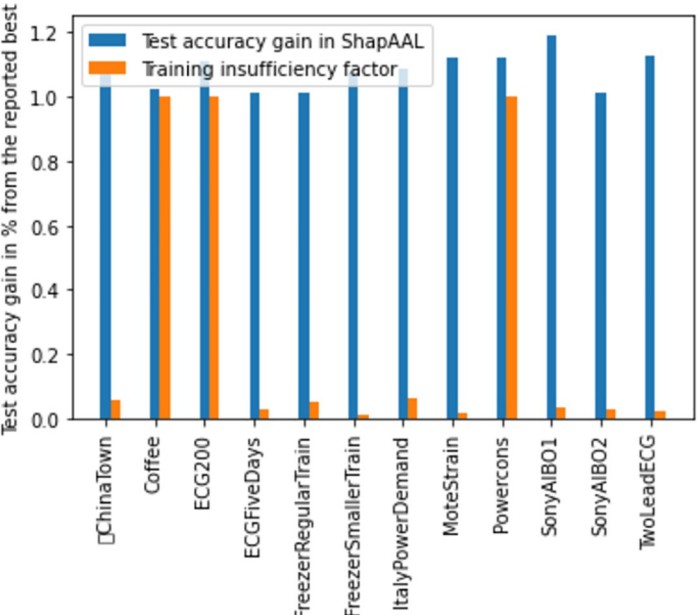

**Fig 16. Empirical support of consistency in learning gain over test data of ShapAAL ($M^{Shapley}$) over base model ($M$) under typical practical constraint of training insufficiency factor $\leq 1$.**

claim that ShapAAL is the apt choice for time series classification tasks under the practical constraint of training data insufficiency. The proposed model attempts to maximize the worst-case classification accuracy owing to the presence of data perturbation, which in philosophy, expands the training space to act as machine generated annotator that creates the possibility of human annotator replacement. Hence, another substantial gain we incur other than better learnability with enriched training process is the avoidance of expensive data labeling processes.

## Discussion

It is well-established in literature with empirical evidences in support of neural scaling law, which hypothesizes that the test error generally decreases as a power law with the number of training data, i.e. more training data is often beneficial for the learnability of a deep learning model and motivated by this neural scaling law, significant investments have been made in data collection [52]. In this work, we have presented our novel ShapAAL algorithm that can potentially overcome the limitation of practical scenarios of insufficiency of training data while performing time series classification tasks including practically important application of cardio-vascular disease detection from ECG recordings. ShapAAL augments the learning method such that unseen training examples are made part of the model learning process along with selection of important training instances through Shapley value computation such that only positively impacting data are included while constructing the computational model. The conventional Shapley value-based feature subset identification relies upon choosing $k$ highest ranking ones [46]. However, aprior knowledge of $k$ is practically infeasible. For instance, the "best" result may be $k = 90\%$ or may be $k = 100\%$ or $k = 60\%$. Hence, the classical approach is not the appropriate choice. Our proposed algorithm is intuitively appealing and principled upon the "Efficiency" and "Null player" Shapley value axioms [46, 47], which is theoretically sound, tractable and practically feasible and supported with empirical investigation as depicted

in Tables 2 and 3. Firstly, we have proposed and validated the unique idea augmentation and ablation of the input features to generate a better learned model. Controlled augmentation of the seen examples to learn better on the unseen examples through introduction of perturbed or virtual data points helps the model to combat the insufficiency in training examples and Shapley-attributed input feature selection refines the input space such that the model gets the opportunity of training more (through augmentation) yet better (Shapley-value based feature ablation). While the augmentation and feature attribution separately improve the test accuracy of the model over different tasks, the combined effect is significant, and it is evident from Tables 2 and 4. The study in Tables 2 and 4 clearly indicates that data augmentation through adversarial learning and subsequent feature space identification for re-learning with appropriate features provide significant impetus to the learning process to learn that compensates the limitation in seen examples and learn appropriately. Secondly, we have provided state-of-the-art comparison of the proposed method and the ShapAAL model with both data augmentation and input attribution features has demonstrated consistently outstanding classification performances over different time series classification tasks, conveniently outperforming the current benchmark and state-of-the-art algorithms as depicted in Tables 3 and 4 and Fig 15.

From a purely pragmatic standpoint, ShapAAL has demonstrated capability of accurately performing diverse set of time series sensor signal classification tasks including identification of time-critical conditions like Myocardial Infarction or heart attack using ECG signal and consistently outperform the state-of-the-art algorithms. Smartphone-based ECG applications are indeed one of the important practical utility of IoT and AI technology [4]. It is known that cardio-vascular diseases are leading cause of human deaths globally [53]. We envisage that the automated ECG analysis is capable of ensuring on-demand, remote monitoring of heart health and can issue accurate alerts when the disease condition is detected with notifying the user and other stakeholders to take relevant clinical actions.

Internet has reached remotest corner of the globe, medical facility is not. We can enable early warning and on-demand automated cardiac care provisioning by leveraging wide-scale deployment of Internet of Things applications acts for developing wireless health monitoring using smartphone and smart ECG sensors like MAX3003 (https://www.maximintegrated.com/en/products/analog/data-converters/analog-front-end-ics/MAX30003.html). It is well-known that early detection and timely intervention can lead to significant life-saving outcomes with substantial reduction of clinical burden. For instance, Myocardial Infarction is to be diagnosed and treated in an urgent manner and an appropriate treatment within first hour can lead to considerable avoidance of deaths and reversal of heart condition. Automated digital screening of cardio-vascular diseases through Internet infrastructure can potentially lead to early detection and in-time screening even at home or at a remote place without real-time access to doctors or cardiologists. Remote screening and monitoring are especially imperative for cardio-vascular disease management. We understand that ShapAAL performs significantly better than the state-of-the-art in cardio-vascular disease detection using ECG signals (for e.g., "ECG200", "ECGFiveDays", "TwoLeadECG" datasets results in Table 3). ShapAAL outperforms the current benchmark in Myocardial Infarction detection with test accuracy of 0.92. ShapAAL as part of the analytic engine for automated detection of Myocardial Infraction condition. The primary objective is to build an early warning and on-demand automated cardiac care provisioning that does not get hindered by the immediate absence of a specialist or the user being in a remote place. As a generic setup, the components of the eco-system can be modularized as applications for user end, medical caregiver end and analytics engine end (where the ECG classification model is hosted. In presence of powerful local machine, smartphone, ECG analysis can be done at edge or locally). Users or the patients install the user end application in his/her smartphone (or it can be installed in a laptop) to proactively interact for

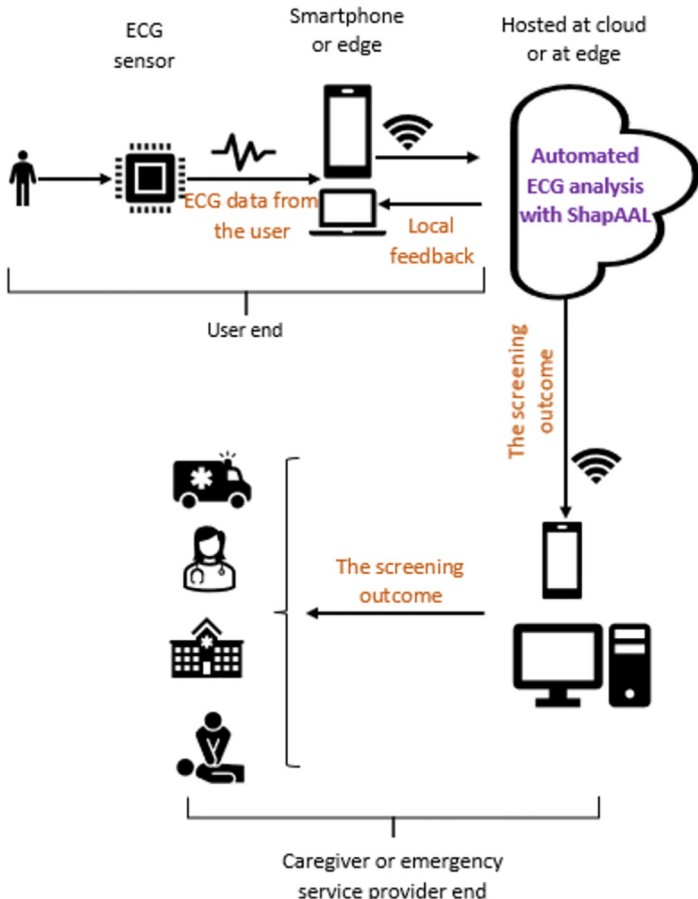

**Fig 17. Early warning, emergency, and on-demand cardiac care provision through automated clinical analytics engine with ShapAAL.**

receiving the cardiac care from the smart healthcare systems with digital therapeutics as part of a typical m-health eco-system. The analytics engine does the job of ECG data interpretation to predict the cardio-vascular disease class. For instance, analytics engine predicts whether the user suffers from Myocardial Infarction condition and sends alerts to the medical caregivers for urgent clinical attention and intervention. The model is trained off-line, and the trained model is deployed on the cloud or at the local workstation as a clinical analytics engine. The on-field ECG data is given as input to the trained model $M^{ShapAAL}$ and the output as one of the disease classes (considering binary or multi-class classification) is considered as the screening outcome. We illustrate the system, which can be potentially developed as an early warning platform for basic CVD screening in Fig 17. Further, we like to mention that clinical screening scenario of the conventional CVD screening and diagnosis need to be changed from a reactive mode to proactive mode. In current conventional setup, users will react when the symptoms flareup. In the most likely scenario, the milder symptoms will be ignored when the clinical facility is far-off. Even the routine check-up, which is necessary for CVD patients may be skipped by the remote patients. Another serious consideration is the missing response of sub-clinical or non-symptomatic condition of CVDs, where the patient might suddenly develop life-threatening conditions. With the proposed automated CVD screening method that can be conveniently performed at home, we expect that the CVD screening will be proactive with

early warning of sub-clinical or non-symptomatic CVDs. We are hopeful that the paradigm shift towards automated basic cardio-vascular disease screening can enable us to achieve the goal of 25% relative reduction in premature mortality due to cardio vascular diseases before 2025 [54].

We like to mention that the ECG-based automated cardio-vascular disease detection as early warning system is illustrated as an example use case scenario. The proposed method is a generic one and would be an ideal choice for different analytics tasks involving the requirement of time series sensor data classification. Another interesting practical application is in food safety and quality assurance ("Coffee" dataset) to identify the type of coffee beans through food spectrographs.

## Conclusion

Our aim of this study is to develop solution for solving the important practical problem of training data scarcity in time series sensor data classification tasks when deploying diverse type of real-world applications including smart cardio-vascular disease detection using ECG data to build effective early-warning, on-demand heart health monitoring eco-system. Our proposed augmented learning with input subset selection approach through Shapley value-based attribution has demonstrated significantly accurate performance over diverse time series sensor data analysis tasks. We have proposed a novel learning mechanism that learns with augmented training to compensate the inadequacy of the training data; unlearns the non-important samples by identifying their contributions to the model predictability through Shapley value computation from coalition game setup with transferable utility; and re-learns with those subset samples. Our novel three-stage time series classification model with learning through augmentation, unlearning the non-contributing input features with Shapley value attribution and finally, relearning through augmentation of selected input features has demonstrated classification efficacy not only through ablation study but also through comparative state-of-the-art investigation. In fact, the intentional introduction of perturbations in the training process of the deep neural network (ResNet) model compels it to learn generalization with crafted and controlled perturbations to create important, unseen input space. The main objective for constructing the learned model when training data is less is to find a way towards minimize the generalization loss over unseen or test or on-filed data. The unique feature of ShapAAL algorithm is the augmentation for learning the unseen data as well as removing the negatively-contributing seen examples in the learning process, which in tandem constitutes superior and effective input space to learn better under training data scarcity problem. Given that Shapley values provide quantitative understanding of fairly attributing the contribution of the input features, the unlearning of detrimental input features has theoretical benefits and we have demonstrated that ablation of such input features has positive impact towards the learnability of the model.

We sincerely hope that the proposed model has the capability to demonstrate practical significance in the development cycle of real-world sensor data classification-based applications including automated prediction of cardio-vascular diseases from physiological marker of heart health like Electrocardiogram to build remote, on-demand smart cardio-vascular health monitoring and early warning system. The proposed method is a generic one for solving time series classification tasks. We envisage that automated analysis with algorithmic screening for cardio-vascular disease identification purpose has the right potential to step towards the long-cherished quest for the availability of a cardio-vascular health management system to intervene for the initial disease screening without expert-in loop.

Our future scope of study includes more exploration towards game theoretic understanding in the construction of a deep learning model with an intuitive rationality perspective of

model's dilemma for prediction over unseen data. The general step for Shapley value computation is using sampling method to estimate the expectation over a distribution of marginals and interpretable machine learning fits to such type of quantified notion of an input feature's contribution. We intend to explore the model interpretability and algorithmic transparency as a future research initiative with model-agnostic interpretability indicating marginal contributions for individual input features. Another interesting idea is to investigate virtual adversarial regularization such that we can consider the perspective of model robustness. While a sophisticated model provides outstanding performance on given dataset, the model may be over-sensitive towards a little adversarial attack. Data augmentation is in fact capable of improving the stability of the model where the model does not have a high confidence at the prediction, but those augmented examples are close to the given seen examples. From practical utility perspective, we shall further focus on introducing prescriptive analytics such that the initial treatment directive can be urgently delivered as a basic critical care, which can be lifesaving as well as provides the emergency caregivers the information to immediately start the basic yet immensely important initial basic clinical procedures. For example, after heart attack, each passing minutes cause more heart tissues to get damaged. When the analytics engine detects heart attack, immediate commencement of medications like aspirins, thrombolytics before a cardiologist's intervention is of immense clinical importance. We intend to bring out a robust remote cardio-vascular management system with automation in the basic screening methods that utilizes the Internet backbone to enable healthcare services to the remotest part of the globe for on-demand screening and basic treatment with both screening and prescriptive functions.

## Supporting information

**S1 Data. Data source.** Experimental datasets are publicly available at https://figshare.com/articles/dataset/Data_zip/21532440/1.
(TXT)

**S1 Table. Hyperparameters.** The hyperparameters used in ShapAAL model construction.
(PDF)

**S1 Fig. Study on the training augmentation control.** We depict the trend of the test accuracy data augmentation control parameter $\alpha$ in ChinaTown dataset by varying $\alpha$ from $0.00 \leq \alpha \leq 0.07$ to understand the response of the model under different strengths of perturbations.
(TIF)

**S2 Fig. Study on the training augmentation control.** We depict the trend of the test accuracy data augmentation control parameter $\alpha$ in Coffee dataset by varying $\alpha$ from $0.00 \leq \alpha \leq 0.07$ to understand the response of the model under different strengths of perturbations.
(TIF)

**S3 Fig. Study on the training augmentation control.** We depict the trend of the test accuracy data augmentation control parameter $\alpha$ in ECG200 dataset by varying $\alpha$ from $0.00 \leq \alpha \leq 0.07$ to understand the response of the model under different strengths of perturbations.
(TIF)

**S4 Fig. Study on the training augmentation control.** We depict the trend of the test accuracy data augmentation control parameter $\alpha$ in ECGFiveDays dataset by varying $\alpha$ from $0.00 \leq \alpha \leq 0.07$ to understand the response of the model under different strengths of perturbations.
(TIF)

**S5 Fig. Study on the training augmentation control.** We depict the trend of the test accuracy data augmentation control parameter $\alpha$ in FreezerRegularTrain dataset by varying $\alpha$ from $0.00 \leq \alpha \leq 0.07$ to understand the response of the model under different strengths of perturbations.
(TIF)

**S6 Fig. Study on the training augmentation control.** We depict the trend of the test accuracy data augmentation control parameter $\alpha$ in FreezerSmallTrain dataset by varying $\alpha$ from $0.00 \leq \alpha \leq 0.07$ to understand the response of the model under different strengths of perturbations.
(TIF)

**S7 Fig. Study on the training augmentation control.** We depict the trend of the test accuracy data augmentation control parameter $\alpha$ in ItalyPowerDemandn dataset by varying $\alpha$ from $0.00 \leq \alpha \leq 0.07$ to understand the response of the model under different strengths of perturbations.
(TIF)

**S8 Fig. Study on the training augmentation control.** We depict the trend of the test accuracy data augmentation control parameter $\alpha$ in MoteStrain dataset by varying $\alpha$ from $0.00 \leq \alpha \leq 0.07$ to understand the response of the model under different strengths of perturbations.
(TIF)

**S9 Fig. Study on the training augmentation control.** We depict the trend of the test accuracy data augmentation control parameter $\alpha$ in PowerCons dataset by varying $\alpha$ from $0.00 \leq \alpha \leq 0.07$ to understand the response of the model under different strengths of perturbations.
(TIF)

**S10 Fig. Study on the training augmentation control.** We depict the trend of the test accuracy data augmentation control parameter $\alpha$ in SonyAIBO1 dataset by varying $\alpha$ from $0.00 \leq \alpha \leq 0.07$ to understand the response of the model under different strengths of perturbations.
(TIF)

**S11 Fig. Study on the training augmentation control.** We depict the trend of the test accuracy data augmentation control parameter $\alpha$ in SonyAIBO2 dataset by varying $\alpha$ from $0.00 \leq \alpha \leq 0.07$ to understand the response of the model under different strengths of perturbations.
(TIF)

**S12 Fig. Study on the training augmentation control.** We depict the trend of the test accuracy data augmentation control parameter $\alpha$ in TwoLeadECG dataset by varying $\alpha$ from $0.00 \leq \alpha \leq 0.07$ to understand the response of the model under different strengths of perturbations.
(TIF)

**S13 Fig. ShapAAL model plot.** We present the complete model description for reproducibility, where the input is "ECG200" training dataset.
(TIF)

## Acknowledgments

Leandro Marin acknowledges the support of PID2020-112675RB-C44 by MCIN/AEI/ 10.13039/5011000011033 for this research work execution.

Antonio J. Jara, Libelium acknowledges the cooperation for data identification and experimentation in QUAFAIR experiment for the Smart and Healthy Ageing through People

Engaging in Supportive Systems: SHAPES—H2020 project (857159) and Comunidad Autonoma de la Region de Murcia (CARM) in HORECOV-21—RIS3MUR FEDER Strengthen research, technological development and innovation.

## Author Contributions

**Conceptualization:** Arijit Ukil, Leandro Marin, Antonio J. Jara.

**Investigation:** Arijit Ukil.

**Methodology:** Arijit Ukil.

**Software:** Arijit Ukil.

**Supervision:** Leandro Marin, Antonio J. Jara.

**Validation:** Leandro Marin, Antonio J. Jara.

**Writing – original draft:** Arijit Ukil.

**Writing – review & editing:** Leandro Marin, Antonio J. Jara.

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
