## [Decision Letter · Decision Letter 0]

14 Oct 2022

PONE-D-22-26084When less is more powerful: Shapley value attributed ablation with augmented learning for practical time series sensor data classificationPLOS ONE

Dear Dr. Ukil,

Thank you for submitting your manuscript to PLOS ONE. After careful consideration, we feel that it has merit but does not fully meet PLOS ONE’s publication criteria as it currently stands. Therefore, we invite you to submit a revised version of the manuscript that addresses the points raised during the review process.

We look forward to receiving your revised manuscript.

Kind regards,

Anand Nayyar, Ph.D.

Academic Editor

PLOS ONE

2. Please note that PLOS ONE has specific guidelines on code sharing for submissions in which author-generated code underpins the findings in the manuscript. In these cases, all author-generated code must be made available without restrictions upon publication of the work. Please review our guidelines at https://journals.plos.org/plosone/s/materials-and-software-sharing#loc-sharing-code and ensure that your code is shared in a way that follows best practice and facilitates reproducibility and reuse. New software must comply with the Open Source Definition.

“Leandro Marin is partially funded by Grant PID2020-112675RB-C44 funded by MCIN 542

(Ministry for Science and Innovation)/AEI (Agencia Estatal de Investigaci´on - State 543

Research Agency)/10.13039/5011000011033.”

 “The work is partially funded by Grant PID2020-112675RB-C44 funded by MCIN (Ministry for Science and Innovation)/AEI (Agencia Estatal de Investigación - State Research Agency)/10.13039/5011000011033. Tata Consultancy Services is funding the work with generous support and entire APC funding.”

Additional Editor Comments :

The Paper needs revisions and is subject for re-review.

Reviewers' comments:

Reviewer's Responses to Questions

**Comments to the Author**

1. Is the manuscript technically sound, and do the data support the conclusions?

Reviewer #1: Yes

Reviewer #2: Partly

2. Has the statistical analysis been performed appropriately and rigorously? 

Reviewer #1: Yes

Reviewer #2: Yes

3. Have the authors made all data underlying the findings in their manuscript fully available?

Reviewer #1: Yes

Reviewer #2: Yes

4. Is the manuscript presented in an intelligible fashion and written in standard English?

Reviewer #1: Yes

Reviewer #2: Yes

5. Review Comments to the Author

Reviewer #1: The authors take up a very important topic, which is time series sensor data classification. Autrozy solidly designed the manuscript. It is very interesting to describe the abstract and explain what the novelty is at work. The authors explain the topic in an interesting way. The strength of the work is a solid literature review, also based on the latest studies and methodology with interesting diagrams and drawings. The weak point is the lack of underlining that the hatch fills the work with. I recommend that you complete this point.

Reviewer #2: Paper Title: When less is more powerful: Shapley value attributed ablation with augmented learning for practical time series sensor data classification

Discusses: Time series sensor data classification tasks often suffer from training data scarcity issue due to the expenses associated with the expert-intervened annotation efforts. For example, Electrocardiogram (ECG) data classification for cardio-vascular disease detection requires expensive labeling procedures with the help of cardiologists. The current state-of-the-art algorithms like deep learning models have shown outstanding performance under the general requirement of availability of large set of training examples. In this paper, we propose Shapley Attributed Ablation with Augmented Learning: ShapAAL, which demonstrates that deep learning algorithm with suitably selected subset of the seen examples or ablating the unimportant ones from the given limited training dataset can ensure consistently better classification performance under augmented training. In ShapAAL, additive perturbed training augments the input space to compensate the scarcity in training examples and Shapley attribution seeks the subset from the augmented training space for better learnability with the goal of better general predictive performance, thanks to the ”efficiency” and ”null player” axioms of transferable utility games upon which Shapley value game is formulated. In ShapAAL, the subset of training examples that contribute positively in a supervised learning setup is derived from the notion of coalition games using Shapley values associated with each of the given examples’ contribution into the model prediction. ShapAAL is a novel push-pull deep architecture where the subset selection through Shapley value attribution pushes the model to lower dimension while augmented training augments the learning capability of the model over unseen data. We perform ablation study to provide the empirical evidence of our claim and we show that proposed ShapAAL method outperforms the current baselines and state-of-the-art results for time series sensor data classification tasks including the practical important ones that detect cardio-vascular diseases from ECG data.

1.Abstract and Conclusion should be concise yet. But should give complete overview of the work and study.

2.Authors can use latest related works from reputed journals like IEEE/ACM Transactions, MDPI, Elsevier, Inderscience, Springer, Taylor & Francis etc and write the references in proper format, from year 2021-2022. Like https://link.springer.com/article/10.1007/s11042-021-11474-y, https://link.springer.com/article/10.1007/s00500-022-06873-8, https://themedicon.com/pdf/engineeringthemes/MCET-02-016.pdf, https://link.springer.com/article/10.1007/s00500-022-07079-8, https://link.springer.com/article/10.1007/s11042-022-12922-z,

https://ieeexplore.ieee.org/abstract/document/9729866/, https://www.sciencedirect.com/science/article/abs/pii/S095741742101472X, https://www.sciencedirect.com/science/article/abs/pii/S1568494621009261

3.The authors seem to disregard or neglect some important finding in results that have been achieved in paper. So, elaborate and explain the results in more details.

4.Improve the results and discussion section in paragraph.

5.Mention the future scope of your present works.

6. PLOS authors have the option to publish the peer review history of their article (what does this mean?). If published, this will include your full peer review and any attached files.

Reviewer #1: No

Reviewer #2: **Yes: **SHUBHAM MAHAJAN

---

## [Author Response · Author response to Decision Letter 0]

26 Oct 2022

Review comments and Response:

5. Review Comments to the Author

Reviewer #1: The authors take up a very important topic, which is time series sensor data classification. Autrozy solidly designed the manuscript. It is very interesting to describe the abstract and explain what the novelty is at work. The authors explain the topic in an interesting way. The strength of the work is a solid literature review, also based on the latest studies and methodology with interesting diagrams and drawings. The weak point is the lack of underlining that the hatch fills the work with. I recommend that you complete this point.

Thanks for the review comment. We have modified the manuscript with careful investigation. The Abstract and Conclusion Sections are modified considerably.

Abstract:

Time series sensor data classification tasks often suffer from training data scarcity issue due to the expenses associated with the expert-intervened annotation efforts. For example, Electrocardiogram (ECG) data classification for cardio-vascular disease (CVD) detection requires expensive labeling procedures with the help of cardiologists. Current state-of-the-art algorithms like deep learning models have shown outstanding performance under the general requirement of availability of large set of training examples. In this paper, we propose Shapley Attributed Ablation with Augmented Learning: ShapAAL, which demonstrates that deep learning algorithm with suitably selected subset of the seen examples or ablating the unimportant ones from the given limited training dataset can ensure consistently better classification performance under augmented training. In ShapAAL, additive perturbed training augments the input space to compensate the scarcity in training examples using Residual Network (ResNet) architecture through perturbation-induced inputs, while Shapley attribution seeks the subset from the augmented training space for better learnability with the goal of better general predictive performance, thanks to the "efficiency" and "null player" axioms of transferable utility games upon which Shapley value game is formulated. In ShapAAL, the subset of training examples that contribute positively to a supervised learning setup is derived from the notion of coalition games using Shapley values associated with each of the given inputs' contribution into the model prediction. ShapAAL is a novel push-pull deep architecture where the subset selection through Shapley value attribution pushes the model to lower dimension while augmented training augments the learning capability of the model over unseen data. We perform ablation study to provide the empirical evidence of our claim and we show that proposed ShapAAL method consistently outperforms the current baselines and state-of-the-art algorithms for time series sensor data classification tasks from publicly available UCR time series archive that includes different practical important problems like detection of CVDs from ECG data.

Conclusion:

Our aim of this study is to develop solution for solving the important practical problem of training data scarcity in time series sensor data classification tasks when deploying diverse type of real-world applications including smart cardio-vascular disease detection using ECG data to build effective early-warning, on-demand heart health monitoring eco-system. Our proposed augmented learning with input subset selection approach through Shapley value-based attribution has demonstrated significantly accurate performance over diverse time series sensor data analysis tasks. We have proposed a novel learning mechanism that learns with augmented training to compensate the inadequacy of the training data; unlearns the non-important samples by identifying their contributions to the model predictability through Shapley value computation from coalition game setup with transferable utility; and re-learns with those subset samples. Our novel three-stage time series classification model with learning through augmentation, unlearning the non-contributing input features with Shapley value attribution and finally, relearning through augmentation of selected input features has demonstrated classification efficacy not only through ablation study but also through comparative state-of-the-art investigation. In fact, the intentional introduction of perturbations in the training process of the deep neural network (ResNet) model compels it to learn generalization with crafted and controlled perturbations to create important, unseen input space. The main objective for constructing the learned model when training data is less is to find a way towards minimize the generalization loss over unseen or test or on-filed data. The unique feature of ShapAAL algorithm is the augmentation for learning the unseen data as well as removing the negatively-contributing seen examples in the learning process, which in tandem constitutes superior and effective input space to learn better under training data scarcity problem. Given that Shapley values provide quantitative understanding of fairly attributing the contribution of the input features, the unlearning of detrimental input features has theoretical benefits and we have demonstrated that ablation of such input features has positive impact towards the learnability of the model. 

We sincerely hope that the proposed model has the capability to demonstrate practical significance in the development cycle of real-world sensor data classification-based applications including automated prediction of cardio-vascular diseases from physiological marker of heart health like Electrocardiogram to build remote, on-demand smart cardio-vascular health monitoring and early warning system. The proposed method is a generic one for solving time series classification tasks. We envisage that automated analysis with algorithmic screening for cardio-vascular disease identification purpose has the right potential to step towards the long-cherished quest for the availability of a cardio-vascular health management system to intervene for the initial disease screening without expert-in loop.

Our future scope of study includes more exploration towards game theoretic understanding in the construction of a deep learning model with an intuitive rationality perspective of model's dilemma for prediction over unseen data. The general step for Shapley value computation is using sampling method to estimate the expectation over a distribution of marginals and interpretable machine learning fits to such type of quantified notion of an input feature's contribution. We intend to explore the model interpretability and algorithmic transparency as a future research initiative with model-agnostic interpretability indicating marginal contributions for individual input features. Another interesting idea is to investigate virtual adversarial regularization such that we can consider the perspective of model robustness. While a sophisticated model provides outstanding performance on given dataset, the model may be over-sensitive towards a little adversarial attack. Data augmentation is in fact capable of improving the stability of the model where the model does not have a high confidence at the prediction, but those augmented examples are close to the given seen examples. From practical utility perspective, we shall further focus on introducing prescriptive analytics such that the initial treatment directive can be urgently delivered as a basic critical care, which can be lifesaving as well as provides the emergency caregivers the information to immediately start the basic yet immensely important initial basic clinical procedures. For example, after heart attack, each passing minutes cause more heart tissues to get damaged. When the analytics engine detects heart attack, immediate commencement of medications like aspirins, thrombolytics before a cardiologist’s intervention is of immense clinical importance. We intend to bring out a robust remote cardio-vascular management system with automation in the basic screening methods that utilizes the Internet backbone to enable healthcare services to the remotest part of the globe for on-demand screening and basic treatment with both screening and prescriptive functions.

We have added some additional results to further substantiate our claim. 

Next, we conduct ablation study to understand the efficacy of the proposed model. An ablation study in general, investigates the performance of a machine learning system by removing few components in order to evaluate the impact of those components in the complete system. Similarly, ShapAAL model construction consists of four components that include the base model (ResNet), Shapley value attribution over the base model, data augmented training on the base model and data augmented training with Shapley attributed feature selection on the base model. We denote $M$ as the base model that is trained with each of the training data, $M^{Shapley}$ as the model that is trained with the training data after discarding the negatively contributing Shapley valued features, $M_{aug}$ is the model that is adversarially trained over over entire augmented training data. $M^{ShapAAL}$ or $M_{aug}^{Shapley}$ is the adversarially trained with the augmented training data with discarding the negatively contributing Shapley valued ones following the deep architecture in Figure ~\\ref{fig862} as depicted in Figure ~\\ref{fig1112}. In Table ~\\ref{table_4}, we depict the "test accuracy" performances of $M$, $M_{aug}$, $M^{Shapley}$ and $M_{aug}^{ShapAAL}$ over the experimental datasets. The ablation study unambiguously indicates that our proposed model $M^{ShapAAL}$ is the superior one. In fact, the trend is also clear that both augmented training and Shapley attributed re-learning have significant positive impact on the learnability of the model, which reflects in the consistent superlative performance of $M^{ShapAAL}$ w.r.t the others. Hence, we establish with the empirical support that less number of input features (Refer Figure ~\\ref{fig2345}) when properly selected can provide better test accuracy. Under training data size constraint scenario, the push-pull architecture of ShapAAL as a coalition game with Shapley attributed push towards lower dimension and concurrently pulling or augmenting the learning capability of the model over unseen data indeed demonstrates significantly improved performance.

Another classical performance merit is the "outperforming" the benchmark. In recent years, number of time series classification algorithms have been proposed in literature, which might not have been updated in the UCR archive repository

\\footnote {available at each of the dataset description URL, for e.g., \\url{https://www.timeseriesclassification.com/description.php?Dataset=ECG200}}. However, we can consider the available benchmark or the best results in the UCR repository of the respective datasets \\footnote{\\url{https://www.timeseriesclassification.com/dataset.php}} as the "reported benchmark". In Figure ~\\ref{fig1d}, we depict the differential test accuracy gain of the algorithms (which has reported results available in public domain) including ShapAAL model w.r.t the reported best results and it is computed as $\\frac{test\\; accuracy\\; of\\; the \\;algorithm\\;-\\; reported\\; benchmark \\;test \\;accuracy}{reported\\; benchmark\\; test\\; accuracy}$ with the aim of being the test accuracy result to be positive, indicating that the concerned algorithm has outperformed the currently reported benchmark result. We observe that proposed ShapAAL steadily outperforms the reported benchmark results in comparison with the relevant benchmark algorithms.

In this manuscript, it is felt that MPCE-score-based classification performance evaluation needs to be part of the main content, which was earlier in the Appendix.

Mean Per-Class Error (MPCE) (~\\cite{wang2017time}) is another useful metric to evaluate the classification performance of the model as: the expected error rate for a single class across each of the test data. For $\\Upsilon$ number of test data with class $\\textit{c}_{\\upsilon}$ and corresponding error rate $err_{\\upsilon}$, we compute MPCE as: $\\frac{1}{\\Upsilon}\\sum \\frac{err_{\\upsilon}}{\\textit{c}_{\\upsilon}}$.\\\\ 

MPCE seems to a more robust as an evaluator of model performance for different datasets of the classes ~\\cite{wang2017time}). Below in Table ~\\ref{tab_76540}, we demonstrate the MPCE results for the ablation study. In MPCE, our aim is to have a lower value, approaching zero.

Another unique feature of the current work is its response to higher number of test instances when it gets trained with smaller number of training examples. We can quantify the learning gain of ShapAAL at the time of testing as: $\\frac{test\\: accurcay_{ShapAAL} \\; - test\\: accurcay_{Base}}{test\\: accurcay_{Base}}$ and also define training insufficiency factor as: $\\frac{Number\\: of\\: training\\: examples}{Number\\: of \\:testing\\: instances}$. In Fig ~\\ref{fig561}, we demonstrate the comparative study of learning gain of ShapAAL on testing data over base model and the insufficiency in the training. We observe that the learning gain of ShapAAL is mostly $\\ge 1$, while training insufficiency factor $\\le 1$. Hence, we further establish our claim that ShapAAL model is the apt choice under practical constraint of training data limitation in solving the time series classification tasks.

We have provided more details in the Discussion Section.

Firstly, we have proposed and validated the unique idea augmentation and ablation of the input features to generate a better learned model. Controlled augmentation of the seen examples to learn better on the unseen examples through introduction of perturbed or virtual data points helps the model to combat the insufficiency in training examples and Shapley-attributed input feature selection refines the input space such that the model gets the opportunity of training more (through augmentation) yet better (Shapley-value based feature ablation). While the augmentation and feature attribution separately improve the test accuracy of the model over different tasks, the combined effect is significant, and it is evident from Table 2 and 4. The study in Table 2 4 clearly indicates that data augmentation through adversarial learning and subsequent feature space identification for re-learning with appropriate features provide significant impetus to the learning process to learn that compensates the limitation in seen examples and learn appropriately. Secondly, we have provided state-of-the-art comparison of the proposed method and the ShapAAL model with both data augmentation and input attribution features has demonstrated consistently outstanding classification performances over different time series classification tasks, conveniently outperforming the current benchmark and state-of-the-art algorithms as depicted in Table 3, Figure 12, and Table 4.

The model is trained off-line, and the trained model is deployed on the cloud or at the local workstation as a clinical analytics engine. The on-field ECG data is given as input to the trained model ShapAAL and the output as one of the disease classes (considering binary or multi-class classification) is considered as the screening outcome. We illustrate the system, which can be potentially developed as an early warning platform for basic CVD screening in Figure 14. Further, we like to mention that clinical screening scenario of the conventional CVD screening and diagnosis need to be changed from a reactive mode to proactive mode. In current conventional setup, users will react when the symptoms flareup. In the most likely scenario, the milder symptoms will be ignored when the clinical facility is far-off. Even the routine check-up, which is necessary for CVD patients may be skipped by the remote patients. Another serious consideration is the missing response of subclinical or non-symptomatic condition of CVDs, where the patient might suddenly develop life-threatening conditions. With the proposed automated CVD screening method that can be conveniently performed at home, we expect that the CVD screening will be proactive with early warning of subclinical or non-symptomatic CVDs.

Reviewer #2: Paper Title: When less is more powerful: Shapley value attributed ablation with augmented learning for practical time series sensor data classification

Discusses: Time series sensor data classification tasks often suffer from training data scarcity issue due to the expenses associated with the expert-intervened annotation efforts. For example, Electrocardiogram (ECG) data classification for cardio-vascular disease detection requires expensive labeling procedures with the help of cardiologists. The current state-of-the-art algorithms like deep learning models have shown outstanding performance under the general requirement of availability of large set of training examples. In this paper, we propose Shapley Attributed Ablation with Augmented Learning: ShapAAL, which demonstrates that deep learning algorithm with suitably selected subset of the seen examples or ablating the unimportant ones from the given limited training dataset can ensure consistently better classification performance under augmented training. In ShapAAL, additive perturbed training augments the input space to compensate the scarcity in training examples and Shapley attribution seeks the subset from the augmented training space for better learnability with the goal of better general predictive performance, thanks to the ”efficiency” and ”null player” axioms of transferable utility games upon which Shapley value game is formulated. In ShapAAL, the subset of training examples that contribute positively in a supervised learning setup is derived from the notion of coalition games using Shapley values associated with each of the given examples’ contribution into the model prediction. ShapAAL is a novel push-pull deep architecture where the subset selection through Shapley value attribution pushes the model to lower dimension while augmented training augments the learning capability of the model over unseen data. We perform ablation study to provide the empirical evidence of our claim and we show that proposed ShapAAL method outperforms the current baselines and state-of-the-art results for time series sensor data classification tasks including the practical important ones that detect cardio-vascular diseases from ECG data.

1.Abstract and Conclusion should be concise yet. But should give complete overview of the work and study.

We have modified the Abstract and Conclusion to provide the overview of the work and study.

Abstract:

Time series sensor data classification tasks often suffer from training data scarcity issue due to the expenses associated with the expert-intervened annotation efforts. For example, Electrocardiogram (ECG) data classification for cardio-vascular disease (CVD) detection requires expensive labeling procedures with the help of cardiologists. Current state-of-the-art algorithms like deep learning models have shown outstanding performance under the general requirement of availability of large set of training examples. In this paper, we propose Shapley Attributed Ablation with Augmented Learning: ShapAAL, which demonstrates that deep learning algorithm with suitably selected subset of the seen examples or ablating the unimportant ones from the given limited training dataset can ensure consistently better classification performance under augmented training. In ShapAAL, additive perturbed training augments the input space to compensate the scarcity in training examples using Residual Network (ResNet) architecture through perturbation-induced inputs, while Shapley attribution seeks the subset from the augmented training space for better learnability with the goal of better general predictive performance, thanks to the "efficiency" and "null player" axioms of transferable utility games upon which Shapley value game is formulated. In ShapAAL, the subset of training examples that contribute positively to a supervised learning setup is derived from the notion of coalition games using Shapley values associated with each of the given inputs' contribution into the model prediction. ShapAAL is a novel push-pull deep architecture where the subset selection through Shapley value attribution pushes the model to lower dimension while augmented training augments the learning capability of the model over unseen data. We perform ablation study to provide the empirical evidence of our claim and we show that proposed ShapAAL method consistently outperforms the current baselines and state-of-the-art algorithms for time series sensor data classification tasks from publicly available UCR time series archive that includes different practical important problems like detection of CVDs from ECG data.

Conclusion:

Our aim of this study is to develop solution for solving the important practical problem of training data scarcity in time series sensor data classification tasks when deploying diverse type of real-world applications including smart cardio-vascular disease detection using ECG data to build effective early-warning, on-demand heart health monitoring eco-system. Our proposed augmented learning with input subset selection approach through Shapley value-based attribution has demonstrated significantly accurate performance over diverse time series sensor data analysis tasks. We have proposed a novel learning mechanism that learns with augmented training to compensate the inadequacy of the training data; unlearns the non-important samples by identifying their contributions to the model predictability through Shapley value computation from coalition game setup with transferable utility; and re-learns with those subset samples. Our novel three-stage time series classification model with learning through augmentation, unlearning the non-contributing input features with Shapley value attribution and finally, relearning through augmentation of selected input features has demonstrated classification efficacy not only through ablation study but also through comparative state-of-the-art investigation. In fact, the intentional introduction of perturbations in the training process of the deep neural network (ResNet) model compels it to learn generalization with crafted and controlled perturbations to create important, unseen input space. The main objective for constructing the learned model when training data is less is to find a way towards minimize the generalization loss over unseen or test or on-filed data. The unique feature of ShapAAL algorithm is the augmentation for learning the unseen data as well as removing the negatively-contributing seen examples in the learning process, which in tandem constitutes superior and effective input space to learn better under training data scarcity problem. Given that Shapley values provide quantitative understanding of fairly attributing the contribution of the input features, the unlearning of detrimental input features has theoretical benefits and we have demonstrated that ablation of such input features has positive impact towards the learnability of the model. 

We sincerely hope that the proposed model has the capability to demonstrate practical significance in the development cycle of real-world sensor data classification-based applications including automated prediction of cardio-vascular diseases from physiological marker of heart health like Electrocardiogram to build remote, on-demand smart cardio-vascular health monitoring and early warning system. The proposed method is a generic one for solving time series classification tasks. We envisage that automated analysis with algorithmic screening for cardio-vascular disease identification purpose has the right potential to step towards the long-cherished quest for the availability of a cardio-vascular health management system to intervene for the initial disease screening without expert-in loop.

Our future scope of study includes more exploration towards game theoretic understanding in the construction of a deep learning model with an intuitive rationality perspective of model's dilemma for prediction over unseen data. The general step for Shapley value computation is using sampling method to estimate the expectation over a distribution of marginals and interpretable machine learning fits to such type of quantified notion of an input feature's contribution. We intend to explore the model interpretability and algorithmic transparency as a future research initiative with model-agnostic interpretability indicating marginal contributions for individual input features. Another interesting idea is to investigate virtual adversarial regularization such that we can consider the perspective of model robustness. While a sophisticated model provides outstanding performance on given dataset, the model may be over-sensitive towards a little adversarial attack. Data augmentation is in fact capable of improving the stability of the model where the model does not have a high confidence at the prediction, but those augmented examples are close to the given seen examples. From practical utility perspective, we shall further focus on introducing prescriptive analytics such that the initial treatment directive can be urgently delivered as a basic critical care, which can be lifesaving as well as provides the emergency caregivers the information to immediately start the basic yet immensely important initial basic clinical procedures. For example, after heart attack, each passing minutes cause more heart tissues to get damaged. When the analytics engine detects heart attack, immediate commencement of medications like aspirins, thrombolytics before a cardiologist’s intervention is of immense clinical importance. We intend to bring out a robust remote cardio-vascular management system with automation in the basic screening methods that utilizes the Internet backbone to enable healthcare services to the remotest part of the globe for on-demand screening and basic treatment with both screening and prescriptive functions.

2.Authors can use latest related works from reputed journals like IEEE/ACM Transactions, MDPI, Elsevier, Inderscience, Springer, Taylor & Francis etc and write the references in proper format, from year 2021-2022. Like https://link.springer.com/article/10.1007/s11042-021-11474-y, https://link.springer.com/article/10.1007/s00500-022-06873-8, https://themedicon.com/pdf/engineeringthemes/MCET-02-016.pdf, https://link.springer.com/article/10.1007/s00500-022-07079-8, https://link.springer.com/article/10.1007/s11042-022-12922-z,

https://ieeexplore.ieee.org/abstract/document/9729866/, https://www.sciencedirect.com/science/article/abs/pii/S095741742101472X, https://www.sciencedirect.com/science/article/abs/pii/S1568494621009261

Thanks for the advice. We have incorporated the latest related works.

In general, machine learning algorithms need to carefully select the supervised 125 features to build a robust model [32]. Optimization method plays an important role in 126 various aspects towards better learned model development under practical constraints 127 [33], [34], [35], [36], [37], [38], [39]. For instance, evolutionary processes with 128 consistent equilibrium for high-quality performance and optimization that achieves 129 quicker convergence is proposed in [35]. It is well-known that the search for global 130 optimization in deep learning algorithms often suffer through spurious local 131 optimization issues. In [36], fusion-based meta-heuristic optimization methods are 132 proposed to solve global optimization tasks.

Added References:

32. Mahajan S, Pandit AK. Hybrid method to supervise feature selection using signal processing and complex algebra techniques. Multimedia Tools and Applications. 2021; p. 1–22. 

33. Mahajan S, Abualigah L, Pandit AK, Altalhi M. Hybrid Aquila optimizer with arithmetic optimization algorithm for global optimization tasks. Soft Computing. 2022;26(10):4863–4881. 

34. Mahajan S, Pandit AK. Image segmentation and optimization techniques: a short overview. Medicon Eng Themes. 2022;2(2):47–49. 

35. Mahajan S, Abualigah L, Pandit AK. Hybrid arithmetic optimization algorithm with hunger games search for global optimization. Multimedia Tools and Applications. 2022; p. 1–24. 

36. Mahajan S, Abualigah L, Pandit AK, Nasar A, Rustom M, Alkhazaleh HA, et al. Fusion of modern meta-heuristic optimization methods using arithmetic optimization algorithm for global optimization tasks. Soft Computing. 2022; p. 1–15.

37. Lakshmi YV, Singh P, Abouhawwash M, Mahajan S, Pandit AK, Ahmed AB. Improved Chan Algorithm Based Optimum UWB Sensor Node Localization Using Hybrid Particle Swarm Optimization. IEEE Access. 2022;10:32546–32565. 

38. Salgotra R, Abouhawwash M, Singh U, Saha S, Mittal N, Mahajan S, et al. Multi-population and dynamic-iterative cuckoo search algorithm for linear antenna array synthesis. Applied Soft Computing. 2021;113:108004. 

39. Singh H, Abouhawwash M, Mittal N, Salgotra R, Mahajan S, Pandit AK. Performance evaluation of Non-Uniform circular antenna array using integrated harmony search with Differential Evolution based Naked Mole Rat algorithm. Expert Systems with Applications. 2022;189:116146.

3.The authors seem to disregard or neglect some important finding in results that have been achieved in paper. So, elaborate and explain the results in more details.

We have elaborated the results to detail out and establish the efficacy of the proposed method. We have made number of additions and modifications in the revised manuscript as depicted below.

{Modification/addition in the Result Section}

Next, we conduct ablation study to understand the efficacy of the proposed model. An ablation study in general, investigates the performance of a machine learning system by removing few components in order to evaluate the impact of those components in the complete system. Similarly, ShapAAL model construction consists of four components that include the base model (ResNet), Shapley value attribution over the base model, data augmented training on the base model and data augmented training with Shapley attributed feature selection on the base model. We denote $M$ as the base model that is trained with each of the training data, $M^{Shapley}$ as the model that is trained with the training data after discarding the negatively contributing Shapley valued features, $M_{aug}$ is the model that is adversarially trained over over entire augmented training data. $M^{ShapAAL}$ or $M_{aug}^{Shapley}$ is the adversarially trained with the augmented training data with discarding the negatively contributing Shapley valued ones following the deep architecture in Figure ~\\ref{fig862} as depicted in Figure ~\\ref{fig1112}. In Table ~\\ref{table_4}, we depict the "test accuracy" performances of $M$, $M_{aug}$, $M^{Shapley}$ and $M_{aug}^{ShapAAL}$ over the experimental datasets. The ablation study unambiguously indicates that our proposed model $M^{ShapAAL}$ is the superior one. In fact, the trend is also clear that both augmented training and Shapley attributed re-learning have significant positive impact on the learnability of the model, which reflects in the consistent superlative performance of $M^{ShapAAL}$ w.r.t the others. Hence, we establish with the empirical support that less number of input features (Refer Figure ~\\ref{fig2345}) when properly selected can provide better test accuracy. Under training data size constraint scenario, the push-pull architecture of ShapAAL as a coalition game with Shapley attributed push towards lower dimension and concurrently pulling or augmenting the learning capability of the model over unseen data indeed demonstrates significantly improved performance.

Another classical performance merit is the "outperforming" the benchmark. In recent years, number of time series classification algorithms have been proposed in literature, which might not have been updated in the UCR archive repository

\\footnote {available at each of the dataset description URL, for e.g., \\url{https://www.timeseriesclassification.com/description.php?Dataset=ECG200}}. However, we can consider the available benchmark or the best results in the UCR repository of the respective datasets \\footnote{\\url{https://www.timeseriesclassification.com/dataset.php}} as the "reported benchmark". In Figure ~\\ref{fig1d}, we depict the differential test accuracy gain of the algorithms (which has reported results available in public domain) including ShapAAL model w.r.t the reported best results and it is computed as $\\frac{test\\; accuracy\\; of\\; the \\;algorithm\\;-\\; reported\\; benchmark \\;test \\;accuracy}{reported\\; benchmark\\; test\\; accuracy}$ with the aim of being the test accuracy result to be positive, indicating that the concerned algorithm has outperformed the currently reported benchmark result. We observe that proposed ShapAAL steadily outperforms the reported benchmark results in comparison with the relevant benchmark algorithms.

In this manuscript, it is felt that MPCE-score-based classification performance evaluation needs to be part of the main content, which was earlier in the Appendix.

Mean Per-Class Error (MPCE) (~\\cite{wang2017time}) is another useful metric to evaluate the classification performance of the model as: the expected error rate for a single class across each of the test data. For $\\Upsilon$ number of test data with class $\\textit{c}_{\\upsilon}$ and corresponding error rate $err_{\\upsilon}$, we compute MPCE as: $\\frac{1}{\\Upsilon}\\sum \\frac{err_{\\upsilon}}{\\textit{c}_{\\upsilon}}$.\\\\ 

MPCE seems to a more robust as an evaluator of model performance for different datasets of the classes ~\\cite{wang2017time}). Below in Table ~\\ref{tab_76540}, we demonstrate the MPCE results for the ablation study. In MPCE, our aim is to have a lower value, approaching zero.

Another unique feature of the current work is its response to higher number of test instances when it gets trained with smaller number of training examples. We can quantify the learning gain of ShapAAL at the time of testing as: $\\frac{test\\: accurcay_{ShapAAL} \\; - test\\: accurcay_{Base}}{test\\: accurcay_{Base}}$ and also define training insufficiency factor as: $\\frac{Number\\: of\\: training\\: examples}{Number\\: of \\:testing\\: instances}$. In Fig ~\\ref{fig561}, we demonstrate the comparative study of learning gain of ShapAAL on testing data over base model and the insufficiency in the training. We observe that the learning gain of ShapAAL is mostly $\\ge 1$, while training insufficiency factor $\\le 1$. Hence, we further establish our claim that ShapAAL model is the apt choice under practical constraint of training data limitation in solving the time series classification tasks.

4.Improve the results and discussion section in paragraph.

We have improved the Results Section with detailed discussion and more results to consolidate our claim as mentioned above. The Discussion Section is also improved with additional details.

{Modification/addition in the Discussion Section}

Firstly, we have proposed and validated the unique idea augmentation and ablation of the input features to generate a better learned model. Controlled augmentation of the seen examples to learn better on the unseen examples through introduction of perturbed or virtual data points helps the model to combat the insufficiency in training examples and Shapley-attributed input feature selection refines the input space such that the model gets the opportunity of training more (through augmentation) yet better (Shapley-value based feature ablation). While the augmentation and feature attribution separately improve the test accuracy of the model over different tasks, the combined effect is significant, and it is evident from Table 2 and 4. The study in Table 2 4 clearly indicates that data augmentation through adversarial learning and subsequent feature space identification for re-learning with appropriate features provide significant impetus to the learning process to learn that compensates the limitation in seen examples and learn appropriately. Secondly, we have provided state-of-the-art comparison of the proposed method and the ShapAAL model with both data augmentation and input attribution features has demonstrated consistently outstanding classification performances over different time series classification tasks, conveniently outperforming the current benchmark and state-of-the-art algorithms as depicted in Table 3, Figure 12, and Table 4.

The model is trained off-line, and the trained model is deployed on the cloud or at the local workstation as a clinical analytics engine. The on-field ECG data is given as input to the trained model ShapAAL and the output as one of the disease classes (considering binary or multi-class classification) is considered as the screening outcome. We illustrate the system, which can be potentially developed as an early warning platform for basic CVD screening in Figure 14. Further, we like to mention that clinical screening scenario of the conventional CVD screening and diagnosis need to be changed from a reactive mode to proactive mode. In current conventional setup, users will react when the symptoms flareup. In the most likely scenario, the milder symptoms will be ignored when the clinical facility is far-off. Even the routine check-up, which is necessary for CVD patients may be skipped by the remote patients. Another serious consideration is the missing response of subclinical or non-symptomatic condition of CVDs, where the patient might suddenly develop life-threatening conditions. With the proposed automated CVD screening method that can be conveniently performed at home, we expect that the CVD screening will be proactive with early warning of subclinical or non-symptomatic CVDs.

5.Mention the future scope of your present works.

The future scope of work is elaborated in the revised manuscript.

Our future scope of study includes more exploration towards game theoretic understanding in the construction of a deep learning model with an intuitive rationality perspective of model's dilemma for prediction over unseen data. The general step for Shapley value computation is using sampling method to estimate the expectation over a distribution of marginals and interpretable machine learning fits to such type of quantified notion of an input feature's contribution. We intend to explore the model interpretability and algorithmic transparency as a future research initiative with model-agnostic interpretability indicating marginal contributions for individual input features. Another interesting idea is to investigate virtual adversarial regularization such that we can consider the perspective of model robustness. While a sophisticated model provides outstanding performance on given dataset, the model may be over-sensitive towards a little adversarial attack. Data augmentation is in fact capable of improving the stability of the model where the model does not have a high confidence at the prediction, but those augmented examples are close to the given seen examples. From practical utility perspective, we shall further focus on introducing prescriptive analytics such that the initial treatment directive can be urgently delivered as a basic critical care, which can be lifesaving as well as provides the emergency caregivers the information to immediately start the basic yet immensely important initial basic clinical procedures. For example, after heart attack, each passing minutes cause more heart tissues to get damaged. When the analytics engine detects heart attack, immediate commencement of medications like aspirins, thrombolytics before a cardiologist’s intervention is of immense clinical importance. We intend to bring out a robust remote cardio-vascular management system with automation in the basic screening methods that utilizes the Internet backbone to enable healthcare services to the remotest part of the globe for on-demand screening and basic treatment with both screening and prescriptive functions.________________________________________

---

## [Decision Letter · Decision Letter 1]

8 Nov 2022

When less is more powerful: Shapley value attributed ablation with augmented learning for practical time series sensor data classification

PONE-D-22-26084R1

Dear Dr. Ukil,

We’re pleased to inform you that your manuscript has been judged scientifically suitable for publication and will be formally accepted for publication once it meets all outstanding technical requirements.

Kind regards,

Anand Nayyar, Ph.D.

Academic Editor

PLOS ONE

Additional Editor Comments (optional):

The Revised Paper stands Accepted with no further revisions.

Reviewers' comments:

Reviewer's Responses to Questions

**Comments to the Author**

1. If the authors have adequately addressed your comments raised in a previous round of review and you feel that this manuscript is now acceptable for publication, you may indicate that here to bypass the “Comments to the Author” section, enter your conflict of interest statement in the “Confidential to Editor” section, and submit your "Accept" recommendation.

Reviewer #1: All comments have been addressed

Reviewer #2: All comments have been addressed

2. Is the manuscript technically sound, and do the data support the conclusions?

Reviewer #1: Yes

Reviewer #2: Yes

3. Has the statistical analysis been performed appropriately and rigorously? 

Reviewer #1: Yes

Reviewer #2: Yes

4. Have the authors made all data underlying the findings in their manuscript fully available?

Reviewer #1: Yes

Reviewer #2: Yes

5. Is the manuscript presented in an intelligible fashion and written in standard English?

Reviewer #1: Yes

Reviewer #2: Yes

6. Review Comments to the Author

Reviewer #1: The manuscript has been adapted to the comments of the reviewer. It is legible, well planned, and substantive.

Reviewer #2: all the comments raised are addressed well the authors

the paper is accepted in present form

i wish the authors best of luck

7. PLOS authors have the option to publish the peer review history of their article (what does this mean?). If published, this will include your full peer review and any attached files.

Reviewer #1: No

Reviewer #2: No

---

## [Editor Report · Acceptance letter]

14 Nov 2022

PONE-D-22-26084R1 

When less is more powerful: Shapley value attributed ablation with augmented learning for practical time series sensor data classification 

Dear Dr. Ukil:

I'm pleased to inform you that your manuscript has been deemed suitable for publication in PLOS ONE. Congratulations! Your manuscript is now with our production department. 

Kind regards, 

on behalf of

Dr. Anand Nayyar 

Academic Editor

PLOS ONE